# Evaluation of convective cloud microphysics in numerical weather prediction model with dual-wavelength polarimetric radar observations: methods and examples

Gregor Köcher[1], Tobias Zinner[1], Christoph Knote[1,3], Eleni Tetoni[2], Florian Ewald[2], and Martin Hagen[2]

[1]Meteorologisches Institut, Ludwig-Maximilians-Universität, Munich, Germany
[2]Deutsches Zentrum für Luft- und Raumfahrt, Institut für Physik der Atmosphäre, Oberpfaffenhofen, Germany
[3]Medizinische Fakultät, Universität Augsburg, Augsburg, Germany

**Correspondence:** Gregor Köcher (gregor.koecher@physik.uni-muenchen.de)

**Abstract.** The representation of cloud microphysical processes contributes substantially to the uncertainty of numerical weather simulations. In part, this is owed to some fundamental knowledge gaps in the underlying processes due to the difficulty to observe them directly. On the path to close these gaps we present a setup for the systematic characterization of differences between numerical weather model and radar observations for convective weather situations. Radar observations are introduced which provide targeted dual-wavelength and polarimetric measurements of convective clouds with the potential to provide more detailed information about hydrometeor shapes and sizes. A convection permitting regional weather model setup is established using 5 different microphysics schemes (double-moment, spectral bin (FSBM), and particle property prediction (P3)). Observations are compared to hindcasts which are created with a polarimetric radar forward simulator for all measurement days. A cell-tracking algorithm applied to radar and model data facilitates comparison on a cell object basis. Statistical comparisons of radar observations and numerical weather model runs are presented on a dataset of 30 convection days. In general, simulations show too few weak and small-scale convective cells. Contoured frequency by altitude diagrams of radar signatures reveal deviations between the schemes and observations in ice and liquid phase. Apart from the P3 scheme, high reflectivities in the ice phase are simulated too frequently. Dual-wavelength signatures demonstrate issues of most schemes to correctly represent ice particle size distributions, producing too large or too dense graupel particles. Comparison of polarimetric radar signatures reveal issues of all schemes except the FSBM to correctly represent rain particle size distributions.

## 1 Introduction

In numerical weather models clouds play an important role by strongly affecting, e.g., the radiation budget or the precipitation formation. Cloud processes are generally divided into two scales: The macrophysics and the microphysics. We refer to "cloud macrophysics" for processes on km scale, namely cloud geometry or cloud coverage, while we refer to "cloud microphysics" for all processes on mm scale or smaller. On coarse-grid weather models, both, macro- and microphysics are unresolved and must be parameterized. Increasing computational power allows numerical weather models to use finer grid spacings, which in turn allow to simulate more and more small-scale processes explicitly. Meanwhile some operational weather models partially

resolve convective updrafts (e.g., Pinto et al., 2015; Baldauf et al., 2011; Seity et al., 2011; Lean et al., 2008, and many more). This progress effectively removes problems arising from cloud macrophysical parameterizations, as they can eventually be solved explicitly. However, this is not the case for microphysical processes due to the large number of hydrometeors present in a cloud. Microphysical processes occur on scales of mm or smaller and are not expected to ever be resolved. As such, the parameterization of microphysics in numerical weather simulations is of increasing relative importance the more the model grid spacing decreases.

Although it is well known that cloud microphysics introduce substantial uncertainty to numerical weather simulations (Li et al., 2009; White et al., 2017; Khain et al., 2015; Xue et al., 2017; Morrison et al., 2020, and many more), the extent of this uncertainty and its underlying reasons remain less clear. Microphysical cloud processes are very complex small scale processes, due to the large variety of shapes, sizes and phases of hydrometeors involved. It is a challenge to represent this complexity correctly in a model since it cannot be resolved explicitly. Instead, the effect of the microphysical processes must be parameterized. This has the potential to introduce uncertainties, as important processes could be misrepresented or missed out completely. In numerical weather models different microphysical schemes of varying complexity exist to parameterize the microphysical processes. Traditionally microphysics schemes are categorized into so-called bulk and bin schemes. Bulk schemes assume a pre-defined shape of the particle size distribution of several hydrometeor classes and predict bulk variables, such as the mass mixing ratio for each of the hydrometeor classes. Depending on the predicted number of variables, the scheme is categorized as a 1-moment (e.g., Kessler, 1969), 2-moment (e.g., Morrison et al., 2009) or even 3-moment scheme (e.g., Milbrandt and Yau, 2005). Bin schemes (e.g., Khain et al., 2004) on the other hand do not assume a pre-defined shape of the particle size distribution but instead use a number of size bins and predict the variables for each of the bins independently. In recent years some alternative schemes have been developed: The Predicted Particle Property (P3) scheme (Morrison and Milbrandt, 2015) deviates from partitioning ice particles into categories of hydrometeor classes with corresponding properties but instead predicts the properties of ice particles, such as the riming mass mixing ratio. Lagrangian cloud models (LCM; e.g., Shima et al., 2009) calculate cloud microphysics based on individual particles (super droplet) that represent a family of particles with the same properties, but this type of scheme will not be covered in the present study.

Several studies have analyzed the performance of cloud microphysics schemes by comparing schemes against each other and against observations (Morrison and Pinto, 2006; Gallus Jr and Pfeifer, 2008; Rajeevan et al., 2010; Jankov et al., 2011; Varble et al., 2014; Fan et al., 2015; Li et al., 2015; Fan et al., 2017; Xue et al., 2017; Han et al., 2019, and many more). However, all of them are limited to case studies. There are some studies that directly use polarimetric radar forward operators to evaluate the performance of cloud microphysics schemes. For instance, Jung et al. (2010) and Snyder et al. (2017) each simulate idealized supercell events to test if the cloud microphysics schemes together with a polarimetric radar forward operator are able to reproduce known supercell radar signatures. Ryzhkov et al. (2011) and Putnam et al. (2017) compare simulated polarimetric radar signals with radar observations to evaluate microphysics schemes, but focus on one or two convective cases. Given the large variability between convective cases, a large number of individual cases is necessary to test if one scheme consistently outperform others in reproducing observations (Flack et al., 2019; Stanford et al., 2019). Few studies have evaluated microphysics schemes on such a statistical basis. Johnson et al. (2015) used a statistical emulation approach to study

the uncertainty produced by several model input parameters but focused on a single idealized convective cloud simulation. Stein et al. (2015) evaluated simulated convective storms over 40 non-consecutive days at varying grid spacings but with only one microphysics scheme. Caine et al. (2013) describe an object-based approach to statistically compare convective cells of a convection-permitting model with radar observations, but they use only two simple microphysics schemes and their statistics are limited to 4.5 days. By comparing two microphysics schemes for different convective events, White et al. (2017) found that the response to cloud droplet number concentrations differs not only between the schemes but also significantly between different convective cases. All of this emphasizes the need for an evaluation of several microphysics schemes over a larger data set on a statistical basis. In an extensive recent overview paper on the challenges in modeling cloud microphysics, Morrison et al. (2020) argue that a rigorous uncertainty quantification on a statistical basis could also help to pinpoint the underlying microphysical processes that cause these uncertainties.

Multiple studies attribute weather simulation errors to poorly constrained cloud microphysics, especially for ice or mixed-phase clouds (e.g., Varble et al., 2014; Stanford et al., 2017). The uncertainty resulting from microphysical cloud processes is in part a result of some fundamental knowledge gaps: It is not well known which processes are poorly represented in numerical models (Morrison et al., 2020). This owes to the difficulty to observe these processes directly. To better constrain the parameters, novel observations are needed to provide corresponding information. These observations must provide information about the key microphysical fingerprints, such as particle properties, their location or ideally conversion rates between hydrometeor classes. Polarimetric radars allow to retrieve hydrometeor classes and shapes and are hence suitable to provide observations of cloud microphysical processes. Kumjian (2012) demonstrate the impact of precipitation processes on polarimetric radar signals, though he focuses mainly on rain processes such as raindrop evaporation or size sorting. Within the framework of Ice-PolCKa (**I**nvestigation of the initiation of **c**onvection and the **e**volution of **p**recipitation using simulati**o**ns and po**l**arimetric radar observations at **C**- and **Ka**-band), a sub-project of the DFG Priority Programme 2115 PROM (Polarimetric Radar Observations meet Atmospheric Modelling - Fusion of Radar Polarimetry and Numerical Atmospheric Modelling Towards an Improved Understanding of Cloud and Precipitation Processes; Trömel et al. (2021)), we exploit the synergy of two polarimetric radars at C- and Ka-band to provide a observational basis for comparison to numerical weather simulations. We evaluate multiple microphysics of different complexity to answer the question: How much complexity is necessary to reproduce polarimetric radar observations?

The goal of this study is to tackle two different aspects:

1. Provide novel observations of cloud microphysics based on dual- wavelength and polarimetric radar measurements using a combination of operational and research-grade radars

2. Evaluate multiple state-of-the-art cloud microphysics schemes for current generation numerical weather prediction models in a common model framework against observations with a large sample size

Eventually, the evaluation should help to identify microphysical processes with obvious differences between radar measurements and weather simulations. However, it is difficult to extract the influence of the cloud microphysics schemes because of

feedbacks between dynamics and microphysics. There are methods that focus on untangling the microphysical impacts from other impacts, e.g., the "piggy backing" method (e.g., Grabowski, 2014). However, operational weather forecast simulations as a whole will always include the feedbacks between microphysics and dynamics as well. Therefore, we decided to use a framework that is applicable to operational weather forecasts and run it over a large number of cases for a statistical comparison, but in this framework we will not be able to perfectly separate the microphysical impacts from possible feedbacks.

We present a setup for the systematic characterization of differences between model simulations with different microphysics schemes and polarimetric radar observations for convective weather situations. This includes the application of a radar forward simulator to the model output and of an automated cell- tracking algorithm to the observations and simulations alike. This allows to objectively compare convective cell characteristics in simulation and observation. We apply this framework to a dataset consisting of 30 days of radar observation and simulations with 5 microphysics schemes of varying complexity.

The potential of the generated data set is demonstrated by showing differences in reflectivity between model and observations in convective clouds to identify issues of microphysics schemes to correctly simulate ice and liquid particle size distributions.

The paper is organized as follows. The methods are described in Sect. 2, which includes our radar data (2.1), the simulation setup (2.2), a description of the microphysical schemes (2.3), the radar forward operator used to bring the model output into radar space (2.4), the cell-tracking algorithm (2.5), and the grid matching of the different radars and the model grid (2.6). In Sect. 3 the microphysics schemes are evaluated by comparing statistics of cloud geometry and frequency (3.1) as well as analyzing frequency diagrams of reflectivity (3.2), polarimetric variables (3.3), and dual-wavelength ratio (3.4) in simulations and observations. In Sect. 4 the results are discussed.

## 2 Data and Methodology

In total, we observed and simulated 30 convective days over 2 years in 2019 and 2020. The majority of these days was spring and summer. For all of them convective precipitation was forecasted. A table listing the dates can be found in Appendix A.

### 2.1 Radar data

The observational data basis is provided by two research radar systems in the area of Munich, Germany, at C- and Ka-band frequencies and a complementary second C-band radar operated by the German Weather Service (DWD; Fig. 1). The C-band research radar Poldirad (Schroth et al., 1988), operated by the German Aerospace Center (DLR), is located in Oberpfaffenhofen southwest of Munich. At 23 km distance the research Ka-band radar Mira-35 is operated by the Meteorological Institute Munich (MIM) of the Ludwig- Maximilians-University (LMU) in the center of Munich. The third radar is an operational C-band radar located in Isen at a distance of 40 km to the Mira-35 radar.

All three radars are polarimetric Doppler radars. Poldirad and the Isen radar are fully polarimetric, sending out electromagnetic waves with horizontal and vertical polarization. Both radars receive the co-polar components backscattered by atmospheric targets. Therefore, polarimetric variables such as differential reflectivity ($Z_{DR}$) or specific differential phase ($K_{DP}$) are available. Poldirad additionally receives the cross-polar components and hence measures the linear depolarization ratio (LDR).

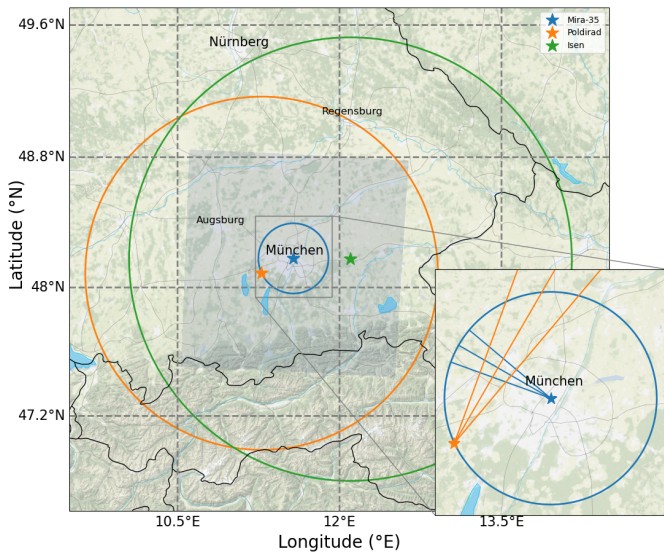

**Figure 1.** Radar locations and model domain. Filled blue area shows the model domain. Stars show the radar locations and the circles show the radar range around each radar. The straight blue and orange lines visualize RHI scans executed by the Mira-35 and Poldirad radar. Background map tiles by Stamen Design (http://stamen.com), distributed under the Creative Commons Attribution (CC BY 3.0) license. Background map data by OpenStreetMap (http://openstreetmap.org; © OpenStreetMap contributors 2021. Distributed under the Open Data Commons Open Database License (ODbL) v1.0.). Roads, rivers and lakes made with Natural Earth (naturalearthdata.com).

The Mira-35 radar is a single-polarization ground-based cloud radar manufactured by METEK GmbH (Bauer-Pfundstein and Görsdorf, 2007). It only transmits horizontally polarized waves but receives co- and cross-polar components. Thus, it possible to measure LDR in addition to the standard reflectivity.

Poldirad and Mira-35 are two research radars without any operational obligations. This allows for synchronized and targeted scan patterns of convective clouds and precipitation on demand. The absolute calibration of reflectivity Z of Poldirad is estimated to have an error of $\pm$ 0.5 dB from calibration with an external electronic calibration device (Reimann, 2013) while the reflectivity error of Mira-35 is estimated to be $\pm$ 1.0 dB (Ewald et al., 2019). We estimate Poldirad $Z_{DR}$ to have an offset of about 0.15 dB from measurements in a liquid cloud layer where $Z_{DR}$ near 0 is to be expected. This offset is corrected before any of the subsequent analysis is done. The Isen radar is part of the DWD operational radar network with a fixed observation strategy. For a complete description of the measurement strategy refer to Helmert et al. (2014). More radar characteristics and configurations can be found in Table 1. This setup allows dual-wavelength and polarimetric measurements of convective clouds and precipitation in the area of Munich.

Two measurement strategies have been applied. For spatial coverage data of the operational DWD Isen radar only is utilized in scan strategy A. The Isen radar is running operationally a volume PPI scan every 5 minutes at 11 elevations from 0.5° to 25° and over the whole azimuth circle of 360°. This provides a good spatial coverage at a high temporal resolution. In Figure 1 the green circle depicts the area that is covered by this strategy.

**Table 1.** Radar characteristics. For the Isen radar the precipitation scan at 1.5° elevation was referenced. For the full configuration of the volume scan, see Helmert et al. (2014).

|  | POLDIRAD | MIRA-35 | ISEN |
|---|---|---|---|
| City | Oberpfaffenhofen | Munich | Isen |
| Location | 48.087 °N, 11.279 °E | 48.148 °N, 11.573 °E | 48.175 °N, 12.102 °E |
| Wavelength | 5.45 cm | 0.85 cm | 5.3 cm |
| Frequency | 5.5 GHz | 35.2 GHz | 5.66 GHz |
| Beamwidth | 1° | 0.6° | 0.9° |
| Range | 120 km | 24 km | 150 km |

In strategy B, Poldirad and Mira-35 are used for coordinated and targeted scan patterns of the same convective cloud. Strategy B starts with a Poldirad overview scan in plan-position indicator (PPI) mode: The elevation angle is kept constant and the azimuth angle is varied. After manually choosing a convective cell from this overview PPI, both radars start to execute three fast scans towards this convective target cloud in the range-height indicator (RHI) scan mode. I.e., the azimuth angle is kept constant while the elevation angle is varied. The first scan is executed exactly towards the direction that was chosen; one is directed to 2° azimuth angle to the left; and one is directed to 2° azimuth to the right. This scan mode is referred to as sector range-height indicator (S-RHI). The 9 intersection profiles resulting from these RHIs give an idea about the variation within the cloud and compensate for potential pointing inaccuracies. In Figure 1 the 6 straight lines (3 orange, 3 blue) visualize these RHI scans. After each S-RHI scan the azimuth direction is adjusted slightly, according to the projected movement of the cell. This cell movement is projected using two previous Poldirad overview PPI scans by calculating the displacement at which the cross-correlation between the two PPI images is at maximum. After a few minutes the S-RHI scans are stopped (manual) and the procedure starts over with another overview PPI scan. This strategy allows targeted dual-wavelength observations of convective clouds in high vertical resolution over a significant fraction of their life-time.

In total we have collected data of strategy B over 5 convective days during summer 2019. The strategy A comprises a larger data set. It consists of the same 5 convective days as well as 25 additional convective days during 2019 and 2020.

## 2.2 Simulation setup

The simulations are performed using the version 4.2 of the Weather Research and Forecasting Model (WRF; Skamarock et al., 2019). Initial and lateral boundary conditions are provided by re-analysis data at 0.25° grid spacing from the Global Forecast System (GFS; NCEP, 2015), available every 6 hours and with hourly forecast data in between. Horizontally, the setup includes a parent Europe domain (3750 km by 3750 km), a two-way nested Germany domain (442 km by 442 km), and a two-way nested Munich domain (144 km by 144 km). The vertical domain extends from the surface to 5 hPa at 40 vertical levels. The nesting ratio is 5:1 with the Europe domain at a horizontal grid spacing of 10 km, the Germany domain at 2 km, and the Munich domain at 400 m. Currently, operational limited area weather models operate at 2 km grid spacing (e.g., 2.8 km

in COSMO-DE of the German Weather Service; Baldauf et al., 2011) which means our inner domain has a resolution that is effectively about 5 times higher and should represent the future of operational limited area weather models most likely including advanced microphysics handling. The Munich domain is centered over the Mira-35 instrument (48.15 °N, 11.57 °E). It covers the Mira-35 range (48 km) and an edge region of an additional 48 km around. All analyses are performed on the innermost Munich domain excluding the edge region, only considering the Mira-35 range (Fig. 1). This area is completely covered by the Poldirad and Isen radar observations. Each simulation consists of 6 hours spin-up and 24 hours simulation time. The spin-up always starts at 18 UTC (20 LST) on the previous day. Thus, the 24 hour forecast exactly covers the day of interest (0 - 24 UTC). The dynamics can freely evolve during the simulation time. The parent Europe domain is nudged towards the global GFS data, by appending a nudging term to the prognostic equations for humidity, temperature and wind that "nudges" the WRF grid value towards the closest GFS grid value for each grid point of the Europe domain above the planetary boundary layer (grid analysis nudging). The inner Germany and Munich domain are not nudged. All days are simulated with 5 different microphysics schemes. Hence, there are 5 simulations available for each of the convective days and the simulation setups only differ in the choice of the microphysics scheme. Other physics options include the Noah Land Surface model (Ek et al., 2003; Chen and Dudhia, 2001), the MYNN2 planetary boundary layer scheme (Mellor-Yamada scheme by Nakanishi and Niino; Nakanishi and Niino, 2006) and the RRTMG radiation scheme (rapid radiative transfer model for general circulation models; Iacono et al., 2008). For any other options, please refer to the WRF namelist that is provided as a supplement to this manuscript.

## 2.3 Description of microphysics schemes

Five different microphysics schemes are employed. Three 2-moment bulk schemes: one from Thompson et al. (2008) (From here on "Thompson") , the one from (Thompson and Eidhammer, 2014) ("Thompson aerosol-aware") and the one from (Morrison et al., 2009) ("Morrison") as well as the "Fast Spectral Bin Microphysics" (FSBM; Shpund et al., 2019) and the "Predicted Particle Properties" scheme (P3; Morrison and Milbrandt, 2015). The FSBM scheme explicitly resolves the particle size distribution (PSD) with a number of bins, while all other schemes generally represent the PSD by a gamma function

$$N(D) = N_0 D^\mu e^{-\lambda D}, \tag{1}$$

where $N_0$ is the intercept parameter, $D$ is the particle maximum diameter, $\mu$ is the shape parameter, and $\lambda$ is the slope parameter. The only exception is snow in the Thompson schemes following a bimodal gamma function as described below.

The mass-size relationships are given by a power-law,

$$m = aD^b, \tag{2}$$

where $m$ is the particle mass and $D$ is the particle diameter. The parameter $a$ and $b$ depend on the hydrometeor class and the scheme used and are described below.

### a. Thompson

The Thompson bulk scheme predicts integral moments of the PSD for five hydrometeor species: cloud ice, cloud water, rain, snow and graupel. Rain and cloud ice are double-moment species which predict mass mixing ratio ($q$) and number concentration ($N$). Snow, graupel and cloud water are single moment, i.e., only the mass mixing ratio is predicted.

The PSDs of rain, cloud ice, graupel and cloud water are represented by gamma distributions (Eq. 1). For rain, graupel and cloud ice $\mu = 0$, i.e., the PSD is an exponential function. Snow and cloud ice have a fixed non-zero $\mu$.

The mass-size relation follows a power-law (Eq. 2). Rain, graupel, cloud ice and cloud water are assumed to be spherical ($b = 3$) with the parameter $a$ depending on the hydrometeor bulk density $\rho$, with

$$a = \rho \frac{\pi}{6}. \tag{3}$$

The bulk density of rain, graupel, cloud ice and cloud water are constant and size independent.

Snow is treated differently in the Thompson scheme compared to other bulk schemes. Instead of the simple gamma function shown in Eq. 1, a bimodal gamma distribution (sum of an exponential and a gamma function) from Field et al. (2005) that is dependent on temperature is used. Snow is not considered to have a constant density across the particle size distribution, the mass is proportional to $D^2$ ($b = 2$) to better fit observations. The parameter $a$ of the mass-size relation is constant at $a = 0.069$.

### b. Thompson aerosol-aware

The Thompson aerosol aware bulk scheme (Thompson and Eidhammer, 2014) is very similar to the older version (Thompson et al., 2008) described in the previous section but includes some changes: While the older version of the Thompson scheme only uses two 2-moment species (rain and cloud ice) and a prescribed number of cloud droplets, the newer version includes activation of aerosols as cloud condensation nuclei (CCN) and ice nuclei (IN). Therefore, it explicitly predicts the droplet number concentration of cloud water and two aerosol variables (CCN and IN).

### c. Morrison

The Morrison bulk scheme predicts integral moments of the PSD for five hydrometeor species: cloud ice, cloud water, rain, snow and graupel. All are double moment species. Particle size distributions follow a general gamma distribution (Eq. 1). Rain, cloud ice, snow and graupel have shape parameter $\mu = 0$, again transforming the particle size distributions into an exponential distribution. For cloud water $\mu$ is a function of droplet number concentration following Martin et al. (1994). All particles are assumed to be spherical with fixed and size-independent bulk densities.

### d. Spectral Bin

In contrast to the bulk schemes, a spectral bin scheme explicitly resolves the PSD by approximation with a number of independent size bins. This has the advantage that no prior assumption about the shape of the PSD is necessary. However, computational costs are much higher, as all microphysical processes are computed for each bin separately. In this study we use

the "Fast Spectral Bin Microphysics" scheme (FSBM; Shpund et al. (2019)) that applies 33 mass-doubling bins, i.e., the mass of the bin $k$ is twice the mass of the bin $k-1$. 5 hydrometeor classes are included: Cloud water, cloud ice, rain, graupel and snow.

*e. Predicted Particle Properties (P3)*

The P3 scheme uses 3 bulk categories: Rain, cloud water and, unlike all the previous schemes, only a single ice category. Instead of predicting mixing ratio and number concentration for multiple ice categories, the P3 scheme predicts properties of this single ice category. Four prognostic ice mixing ratio variables are predicted: total ice mass, rime mass, rime volume and number mixing ratio. Based on these variables more properties are derived, such as rime mass fraction $F_r$ (Ratio of rime mass and ice mass mixing ratio) or rime density $\rho_r$ (Ratio of rime mass and rime volume mixing ratio). All particle size distributions follow a general gamma distribution (Eq. 1). For cloud droplets the shape parameter $\mu$ follows observations of Martin et al. (1994). For rain $\mu$ follows observations of Cao et al. (2008). For ice $\mu$ follows observations of Heymsfield (2003).

Mass-size relationships follow a power law (Eq. 2). The parameter $a$ and $b$ depend on the size of the ice. The scheme distinguishes between small ice, unrimed ice, partially rimed ice and fully rimed ice (graupel/hail). Small ice and graupel are considered spherical ($b=3$) with parameter $a$ given by Eq. 3, where the ice bulk density $\rho$ equals $917\,\mathrm{kg\,m^{-3}}$ for small ice and varies for graupel/hail. Unrimed ice, grown by vapor diffusion or aggregation, and partially rimed ice have an effective density that is generally less than that of an ice sphere ($b=1.9$). The parameter $a$ follows an empirical relationship from Brown and Francis (1995) ($a = 0.0121\,\mathrm{kg\,m^{-b}}$) for unrimed ice and depends on the rime mass fraction $F_r$ for partially rimed ice ($a = 0.0121/(1-F_r)\,\mathrm{kg\,m^{-b}}$), i.e., $a$ increases with the rime mass fraction. Rain and cloud water are considered spherical with $b=3$ and $a$ following Eq. (3) and a bulk density $\rho$ of $1000\,\mathrm{kg\,m^{-3}}$.

## 2.4 Radar forward operator

To compare the WRF model output against radar observations, version 3.33 of the Cloud Resolving Model Radar Simulator (CR-SIM; Oue et al., 2020) is used. CR-SIM is based on the T-matrix method to compute the scattering characteristics of hydrometeors and is able to simulate polarimetric and Doppler radar variables for several radar frequencies, including C-band and Ka-band that are used in this study. The variables include, among many others, the reflectivity (Z) and specific attenuation (A) at vertical and horizontal polarization, differential reflectivity ($Z_{\mathrm{DR}}$) and specific differential attenuation (ADP). Given that CR-SIM supports both C- and Ka-band frequencies, we are also able to simulate the dual-wavelength ratio (DWR) by performing the forward simulation for the C-band radar as well as the Ka-band radar. The dielectric constant of water is 0.92. Solid phase hydrometeors are assumed to be dielectric dry oblate spheroids and are represented as air in an ice matrix. The refractive index hence depends on the hydrometeor density and is computed using the Maxwell Garnett (1904) mixing formula. There are no mixed phased particles simulated. This means mixed phase radar signatures (for example the "bright band", Austin and Bemis, 1950) will not be reproduced by the simulation. In order to simulate polarimetric radar observables, a radar forward simulator must assume particle shapes and particle orientation. The particle orientation assumptions are the same for all schemes. It is assumed that the particle orientations are 2D Gaussian distributed with zero mean canting angle as in Ryzhkov et al. (2011). The width of the angle distributions is specified for each hydrometeor class: 10° for cloud, rain, and ice

and 40° for snow, unrimed ice, partially rimed ice, and graupel. Regarding the shape assumptions, cloud droplets are simulated as spherical (aspect ratio of 1) and raindrops are simulated as oblate spheriods with a changing axis ratio dependent on the drop size according to Brandes et al. (2002) in all schemes. For ice hydrometeor classes, the same aspect ratio assumptions are applied for all schemes except the P3 scheme: cloud ice is assumed as oblate with a fixed aspect ratio of 0.2. Snow is assumed as oblate with a fixed aspect ratio of 0.6. Graupel is assumed to be oblate with an aspect ratio that is changing from 0.8 to 1, dependent on the diameter and according to Ryzhkov et al. (2011):

$$ar = 1.0 - 0.02 \quad \text{if } D < 10 \text{ mm},$$
$$ar = 0.8 \quad\quad\quad \text{if } D > 10 \text{ mm}.$$

The P3 scheme does not provide the standard ice hydrometeor classes. Instead, the aspect ratio of small ice (spherical, fixed aspect ratio of 1), unrimed ice (oblate, fixed aspect ratio of 0.6), partially rimed ice (oblate, fixed aspect ratio of 0.6) and graupel (spherical, fixed aspect ratio of 1) is assumed by CR-SIM. This means in comparison to the other schemes that the P3 simulation deviates for small ice (aspect ratio of 1 in P3, while cloud ice in other schemes is assumed to have an aspect ratio of 0.2) and graupel (0.8 - 1 in other schemes, while graupel particles in P3 are assumed to have an aspect ratio of 1). Resulting differences in the radar signal are discussed in the result section 3 whenever it might influence the simulated radar signal.

## 2.5 Cell-Tracking

This study focuses on convective clouds and precipitation. To identify and track convective cells in simulations and observations, the open source python package TINT (TINT is not TITAN; Fridlind et al., 2019) is used. TINT is based on the Thunderstorm Identification, Tracking, Analysis and Nowcasting package (TITAN; Dixon and Wiener, 1993). Convective cells are identified using minimum thresholds for reflectivity (32 dBZ) and cell area (8 km$^2$). 32 dBZ is at a common magnitude to identify convective storms (e.g., Dixon and Wiener, 1993; Jung and Lee, 2015). Higher thresholds potentially miss moderate or weaker convective cells, while lower thresholds will misidentify more non-convective echos as convective cells. A cell motion vector is found by calculating cross-correlation of the reflectivity field in the cell neighborhood of two subsequent time steps and a correction based on prior cell movement. Possible convective cell pairs are compared and matched using an algorithm from TITAN that uses a cost function combining travel distance and volume change of the possible cell pairs. The cell-tracking is applied to simulated and observed reflectivity of the Isen radar only. The simulated and observed reflectivity from Mira-35 and Poldirad is not used for cell-tracking. This way we ensure to have one unique definition to locate convective cells and prevent varying cell definitions depending on the radar that is simulated. More detailed information about TINT can be found in Fridlind et al. (2019) and Dixon and Wiener (1993). TINT does not deal with splits (one cell splits into multiple cells) or mergers (multiple cells merge into one cell), but it was specifically designed for tracking of convective cells over large datasets and is straightforward to apply to our data (Fig. 2).

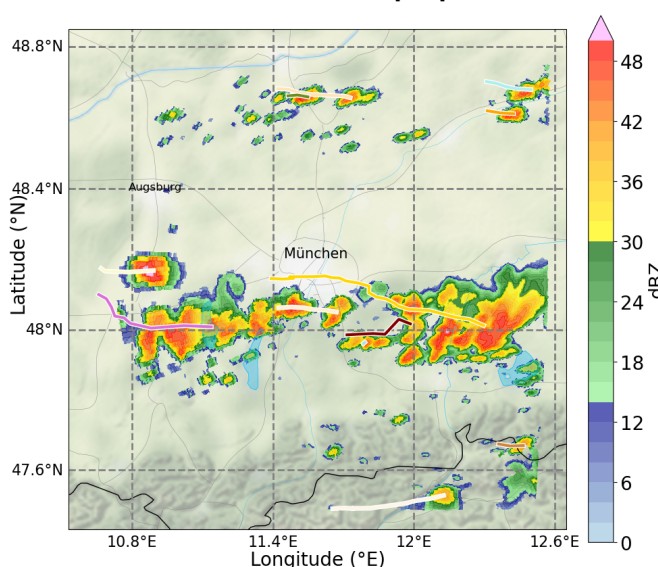

**Figure 2.** Example of cell-tracking with TINT: Colored background is the reflectivity simulated with WRF and CR-SIM, solid lines and numbers represent the TINT tracks and TINT cell identifier.

## 2.6 Grid Matching and attenuation correction

Radar data and model output are available on different grids. To allow for a comparison these grids must be matched first. In a first step, the model data is transformed to a spherical grid of the corresponding radar. For example, simulated Mira-35 radar data is transformed to a spherical grid with a range resolution of about 31 m and a maximum range of 24 km. The transformation utilizes the source code radar_filter, which is available on the website of the Stonybrook University together with the CR-SIM source code (https://you.stonybrook.edu/radar/research/radar-simulators/; last accessed 21.09.2021). The radar_filter considers

beam propagation effects. I.e., for the interpolation to a grid point of the target spherical grid, all Cartesian input grid points that are within the beam width are included with a weight depending on the distance to the radar volume center. If no Cartesian grid point falls into the radar beam, the nearest grid point is used. In the next step attenuation correction is applied along the beam. The correction is applied by subtracting the accumulated (along the range coordinate) simulated attenuation from the uncorrected reflectivity

$$Z_{\text{corr,r}} = Z_{\text{r}} - 2 \cdot \Delta r \cdot \sum_{i=0}^{i=r} A_{\text{i}}. \tag{4}$$

Here the simulated reflectivity without attenuation correction at range gate $r$ is given by $Z_{\text{r}}$. $A_{\text{i}}$ is the simulated attenuation in dB/m at range gate $i$, and $\Delta r$ is the radar range resolution in m. The factor 2 takes into account the fact that the beam travels

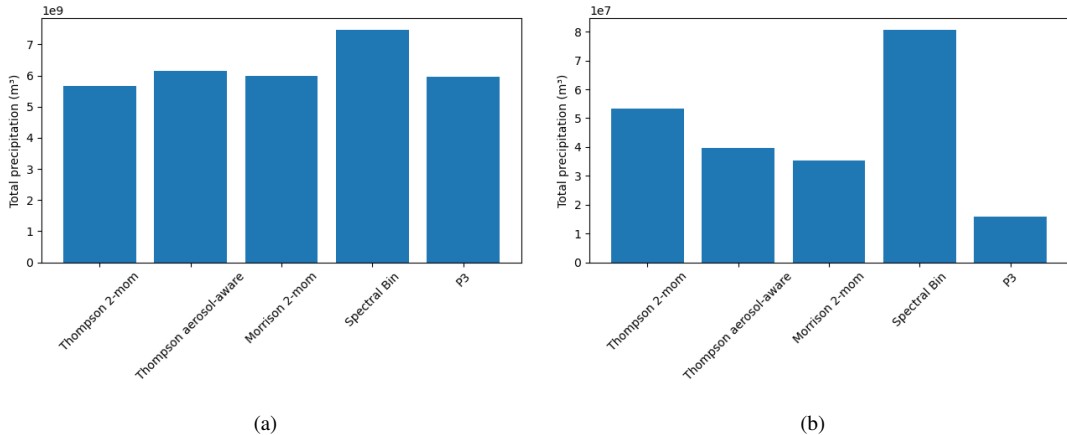

(a)                                    (b)

**Figure 3.** Total simulated precipitation over a) 30 days and b) on 01.08.2020 over the Munich area.

twice through each grid box (from antenna to target and back). In the same way, the differential reflectivity $Z_{DR}$ is corrected with the simulated differential attenuation ADP.

In a last step, all data (model and radar) is transformed back to a Cartesian grid that exactly covers the Munich domain of the model (144 km by 144 km) with a 400 m by 400 m by 100 m (vertical) grid spacing. This is done by applying a nearest neighbor interpolation that chooses the closest radar bin for each of the Cartesian grid point. Only grid boxes within the lowest and highest radar beam are considered. All grid boxes below the lowest or above the highest beam are masked out. Then, the cell-tracking with TINT is applied to this Cartesian grid in exactly the same way for model and radar data, by passing Py-ART

grid objects (Helmus and Collis, 2016) created from the Cartesian grid data to TINT.

## 3   Comparison of model and radar observations

An example for the impact of the microphysics scheme choice is given in Figure 3. It shows the total accumulated precipitation over the Munich area as simulated by WRF simulations while only varying the microphysics scheme. Overall, the total accumulated precipitation over the whole domain and over longer periods (3a; 30 days) is similar between all schemes except the

315 bin scheme. However, the deviations can be much larger during single days (e.g., 3b; 1st of August 2020). The total precipitation over all surface grids varies between all 5 schemes, in this case by more than $6 \cdot 10^7 \, \mathrm{m}^3$ between the P3 and the bin scheme. This illustrates the variation between simulations as a result of the choice of microphysics scheme alone. In the following part, we analyze the resulting deviations in more detail.

### 3.1   Cloud geometry and frequency

We begin our comparison with an evaluation of the geometric properties of simulated and observed clouds. Figure 4 shows histograms of the convective cell core extent (altitude of continuous 32 dBZ volumes) as well as the maximum cell reflectivity

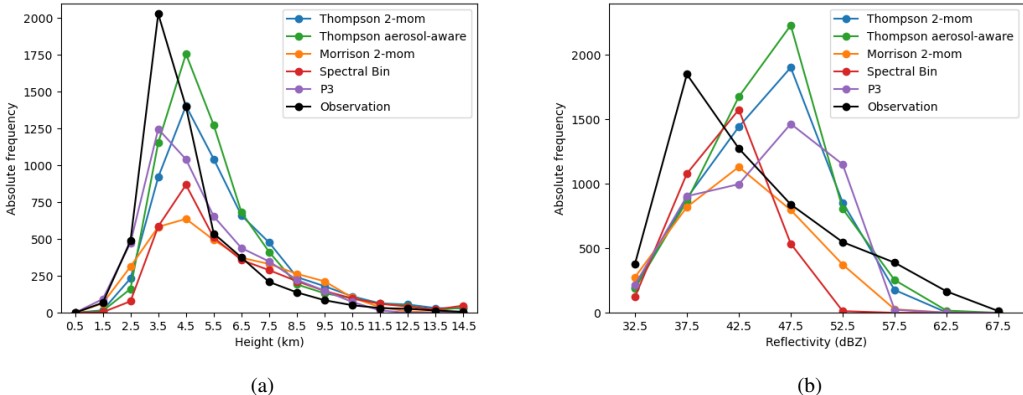

(a)  (b)

**Figure 4.** Cell core height (a) and cell maximum reflectivity (b) distribution for inner simulation domain over 30 convection days for observations and 5 microphysics scheme simulations.

provided by strategy A. At each 5-min time step during 30 convective weather days all cell detections are summed up on DWD Isen observation data or CR-SIM forward simulations. This means, this analysis is independent of possible matching errors of the cell-tracking, as the identified convective cells at each time step are counted independently. E.g., a single cell detected for 30 min would contribute to the statistics six times. The cell core top heights of observed cells (Fig. 4a) show a distinct peak with more than 2000 cell detections at an altitude of 3-4 km. This corresponds to about 40 % of all cell detections by the radar. All NWP simulations independent of the microphysics scheme are able to reproduce a peak at a similar altitude but none of them as pronounced as in the observations. The two Thompson schemes show a tendency towards slightly higher cell core heights of 4-5 km. Reflectivities of more than 32 dBZ above the melting layer are mostly related to big graupel particles in our simulations, and to a lesser extent rain likely lifted by updrafts. Especially the Thompson schemes more frequently simulate graupel particles that produce very high reflectivities of more than 45 dBZ above the melting layer (see Appendix B).

A similar approach to compare cloud geometry in simulation and radar observation was followed in Caine et al. (2013). They objectively compare simulated cell characteristics with observations over 4.5 days after applying a cell-tracking algorithm on their data. Among other things, they found the simulated convective cells to reach higher altitudes on average compared to their radar observations, which is also visible in our analysis. This is independent of the chosen cloud microphysics scheme and mainly a result of the missing small-scale cells in the simulations which is indicative of a resolution effect: the very small cell heights correspond to small cells that we might not be able to resolve even with our 400 m grid spacing.

Regarding the total number of cell detections the Thompson schemes are closest to the observed number. 5458 cell detections are counted, i.e., the number of cells in all 5-minute observation time steps. The Thompson-aerosol aware scheme (6035) is still close to the observed number; the basic Thompson scheme (5468) is the closest; and the P3 (4758) has fewer cells. Especially the Morrison- (3427) and FSBM-scheme (3326) produce too few convective cells. This difference is mainly a result of missing small-scale development (early stages, weak cells) in the simulations. For fully developed thunderstorms (cell core top heights

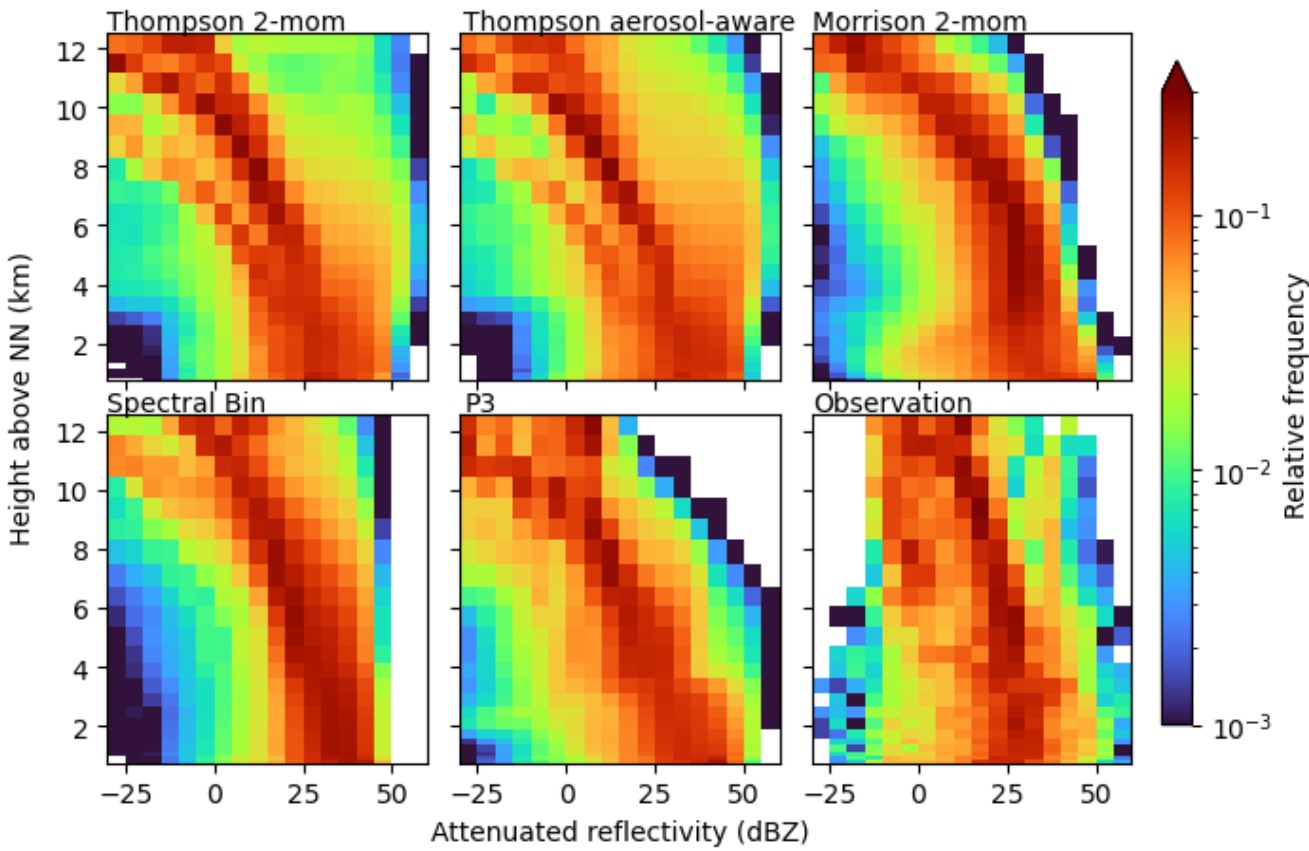

**Figure 5.** CFADs of simulated and measured reflectivity over 5 convective days in 2019. Radar observations with Poldirad.

> 7 km) all schemes produce numbers that are slightly larger than in the observations (observations: 554, Thompson: 1139, Thompson aerosol-aware: 948, Morrison: 928, FSBM: 899, P3: 780). The related distribution of maximum reflectivity of each
cell provides some clarification (Fig. 4b). The observed high occurrence of weaker cells is only partially visible in FSBM and Morrison schemes. While the total number of weaker cells (max cell reflectivity at 35 - 40 dBZ) is still too low, the Morrison and FSBM schemes show the highest relative occurrences for relatively weak cells between 40 - 45 dBZ maximum reflectivity. This does still not represent the pronounced peak of observed cells at weaker reflectivities of 35 - 40 dBZ well. The other three schemes produce too many medium intensity cells and too few low intensity cells. At the other end of the reflectivity spectrum,
none of the models is able to reproduce the occurrence of the strongest reflectivities at more than 57 dBZ. In part, this is most likely related to numerical smoothing of local and rare values in the NWP model.

## 3.2 Profiles of Reflectivity

Contoured frequency by altitude diagrams (CFADs; Yuter and Houze Jr, 1995) for reflectivity of observed and simulated convective cells are shown in Fig. 5 provided by scan strategy B. This scan strategy provides dual-frequency profiles of high vertical resolution through convective clouds. The radar observation CFADs contain about 1300 profiles in convective clouds. The simulated CFADs consist of many more profiles (on the order of $10^5$), because (1) all cells present during one time step on the model domain are analyzed and (2) all columns within each identified cell are included (opposed to the 3x3 profiles that an S-RHI observation provides). The restriction to the center profile of the convective cell, which is a default output of the TINT cell-tracking, would have been an alternative approach. We decided against it for three reasons: (1) the observation was targeted at the location of highest reflectivity and the geometric TINT cell center is not necessarily the location of highest reflectivity; (2) using the S-RHI strategy we include more variation from each cell compared to one center profile; (3) more profiles provide a better statistical basis for intercomparison of schemes. In Figure 5 the simulated reflectivities are corrected for attenuation to make them comparable to the radar observations. Below the melting layer high reflectivities of more than 30 dBZ up to 45 dBZ are simulated most frequently. Overall the schemes agree in the simulated reflectivity in this area mostly caused by rain and graupel. They differ only in the spread. The Morrison scheme shows a higher spread, more often simulating reflectivities below 30 dBZ and even down to 0 dBZ. In contrast, the FSBM produces reflectivities below 25 dBZ less often than the others within the convective cells. Compared to the observed CFAD, high reflectivities below the melting layer are generally modelled too frequently. This is in agreement with Putnam et al. (2017) who compare radar signals simulated by 5 different microphysic schemes for two case studies and find that especially the Morrison scheme but to a lesser extent also the Thompson scheme produces too high Z. They attribute this to stratiform rain PSDs that contain too many large drops, to an overforecast of the precipitation coverage overall and in case of Morrison, to a high bias of wet graupel in convective regions. Given that the forward simulator applied in this study does not consider wet particles, we find the high bias in Z exists even without considering wet graupel and comes mostly from rain, suggesting PSDs that contain too many large rain drops compared to the observations.

Above the melting layer simulated reflectivities start to decrease with height. This is a fingerprint of ice growth processes where falling particles increase in size by deposition, aggregation or riming. At these subfreezing heights the schemes show more deviations from each other. While most schemes exhibit a smooth transition from ice to liquid phase, the prominent exception is the P3 scheme for which reflectivities abruptly increase by about 15 dBZ at the melting layer height (approximately at 3.6 km height, varies among cases). All other schemes show a slow and smooth increase in reflectivity, which better agrees with our observations. However, given that the reflectivity within rain was too high, the reflectivity distribution above the melting layer height is reproduced quite well by the P3 scheme. Most other schemes directly above the melting layer height extend to higher reflectivities, showing reflectivities greater than 25 dBZ too often. The Thompson schemes even simulate reflectivities of more than 45 dBZ above the melting layer height frequently. These extreme reflectivity values are produced mostly by graupel and to lesser extent by rain (see Appendix B for CFADs of radar signals separated by hydrometeor classes). Compared to our measurements these reflectivities are unrealistically large. A high bias in reflectivity could be produced in

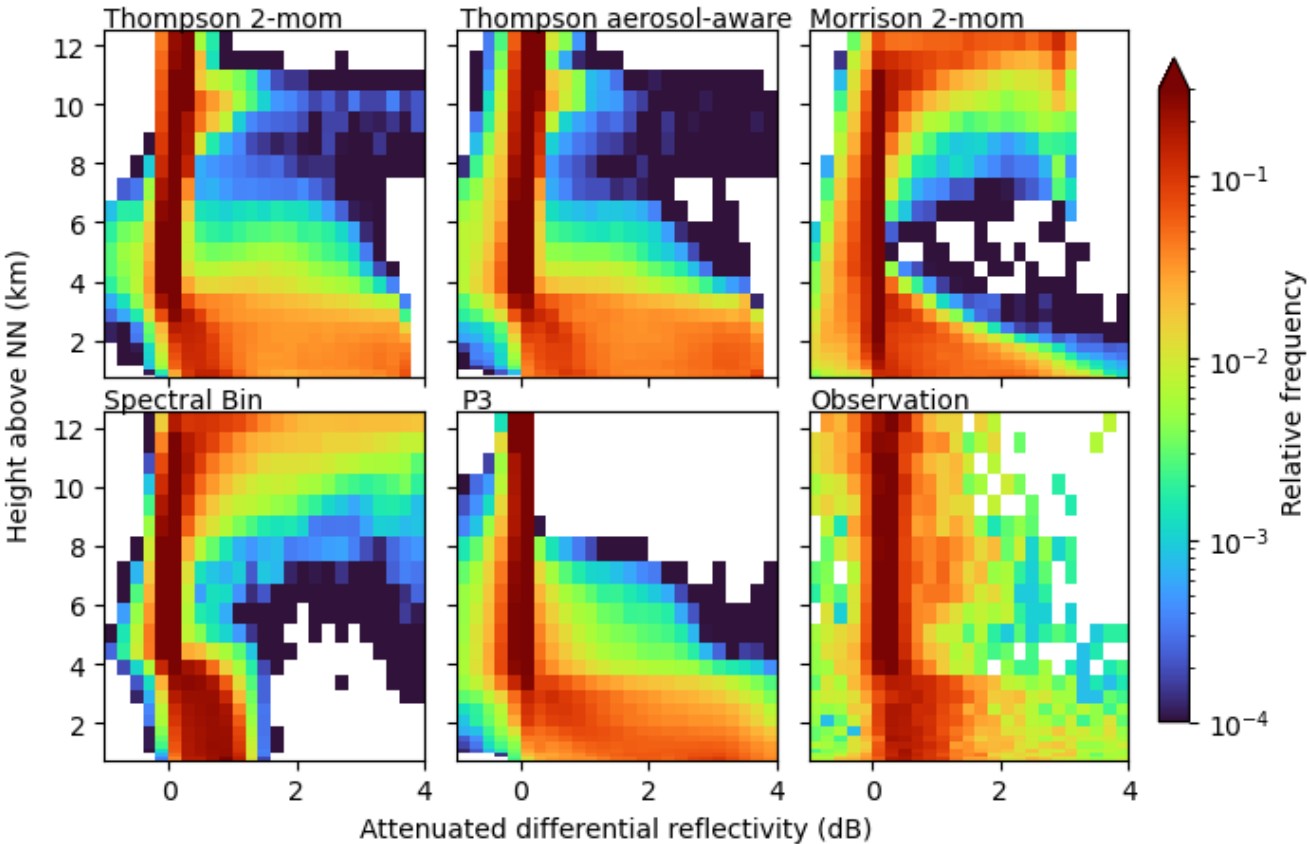

**Figure 6.** CFADs of simulated and measured differential reflectivity $Z_{DR}$ over 5 convective days in 2019. Observation with the Poldirad radar.

principle by three mechanisms: the simulated particles are 1) too dense, 2) too many, or 3) too large. The graupel densities assumed by the schemes (and correspondingly in the forward simulator) are 500 $kg\,m^{-3}$ in the Thompson schemes and 400 $kg\,m^{-3}$ in the Morrison and FSBM scheme. The higher graupel density could explain the higher bias seen in the Thompson scheme compared to the moderate bias in Morrison and FSBM, but the underlying PSD could also play a role. Reflectivity overestimation in deep convection at subfreezing temperatures was found by other studies as well (e.g., Stanford et al., 2017; Varble et al., 2011) and is explained to be a result of too large graupel or snow particles, likely a product of overly strong updrafts. Stanford et al. (2017) show that this bias not only exists for bulk schemes, but also for a bin scheme. We can confirm that with our simulations: the bias exists for the FSBM scheme too, even though it appears to be strongest in the Thompson scheme. However, this could be a consequence of the assumed higher assumed graupel density.

 ## 3.3 Profiles of polarimetric variables

The same analysis is possible for simulated and observed polarimetric variables, e.g., differential reflectivity $Z_{DR}$ (Fig. 6). We found $K_{DP}$ to provide not much additional value, in part due to noisy observations, which is why we neglect $K_{DP}$ in the subsequent analysis. Strong differences between the simulations are visible in the liquid phase below the melting layer. While most schemes show a wide spread over the whole range of 0-4 dB within rain, the FSBM only produces $Z_{DR}$ values up to around
1.5 dB. This is in much better agreement with the observations where $Z_{DR}$ values of up to 1.5 have been measured most of the time, though also covering slightly higher $Z_{DR}$. Here, the advantage of the FSBM that uses a discrete PSD becomes apparent. The FSBM model is able to explicitly predict rain droplets of each bin which is more flexible and potentially better captures the variability in observed PSDs (better size-sorting). Ryzhkov et al. (2011) for example evaluate radar signals simulated from a spectral bin scheme for a hailstorm case and find that their spectral bin scheme produces PSDs for rain that deviate from the
gamma distribution. Bulk schemes would not be able to reproduce these PSDs and since radar signals strongly depend on the PSD, Ryzhkov et al. (2011) argue that spectral bin schemes are better suited to simulate polarimetric radar signals. However, contributions by other microphysical processes, such as drop breakup or evaporation could also facilitate the ZDR signatures and were not examined in this study. All other schemes use a gamma distribution (Eq. 1) with a shape parameter $\mu = 0$ for rain. This effectively is an exponential (Marshall-Palmer) PSD which has a slope that is too weak: There are too few small
rain droplets and too many big droplets. Putnam et al. (2017) find similar results regarding $Z_{DR}$ signatures near the surface: in their two case studies, the simulations with Thompson and with Morrison cloud microphysics showed incorrect $Z_{DR}$ maxima associated with isolated large drops at locations of weak convection where this would not be expected. All of this suggests that the underlying rain particle size distributions are better captured by the FSBM compared to the bulk schemes.

In order to separate the analysis into reasons due to differences in the underlying modelled microphysics and due to different
processing in the forward simulator, we examined rain particle size distributions directly produced by the NWP model (Rain PSD CFAD in Appendix B). The FSBM scheme provides the drop size distributions over a number of size bins, for the bulk schemes we calculated the distributions according to the schemes parameterization. The FSBM bins are approximated by calculating the number of droplets for the geometric center of the FSBM bin. The calculated number of droplets for the given bin center diameters are then summed up over all time steps and over the grid boxes at each height and visualized as a relative
frequency. Only grid boxes that were flagged as a convective cell by the TINT cell tracking are considered. The rain PSD CFAD confirms the findings of the $Z_{DR}$ CFAD: the two Thompson schemes simulate large rain drops from the surface up to the melting layer height and even above, while the Morrison scheme produces large rain drops only at the surface and the FSBM produce the highest frequency of small drops.

Directly above the melting layer the FSBM and Morrison schemes show $Z_{DR}$ values close to 0, while the P3 and the Thomp-
son schemes have their frequency maximum at 0, but show more spread also to higher $Z_{DR}$ values. $Z_{DR}$ of 0 is associated with spherical particles. The signal directly above the melting layer height is generally dominated by graupel, which has the highest reflectivity signal (see Appendix B for separation by hydrometeor class) and is associated with $Z_{DR}$ values of 0, due to the assumed aspect ratio of 1 in the forward simulation. The sparse but large values of $Z_{DR}$ in the two Thompson and the P3

scheme are predominantly caused by rain, likely lifted by strong updrafts in the convective situations. The FSBM scheme also
shows $Z_{DR}$ signals originating from rain particles in that area, but the total $Z_{DR}$ is reduced by a significant contribution from other hydrometeors with a lower $Z_{DR}$. Only the Morrison scheme shows no contribution by lifted rain drops directly above the melting layer. The observations show a little more spread compared to Morrison and FSBM in that area, but the $Z_{DR}$ does not reach values as high as for the Thompson and P3 schemes. There are multiple possible explanations for the differences to the observations: Compared to the observed convective cells, 1) more (less) large rain drops are lifted above the melting layer
height in Thompson/P3 (Morrison/FSBM), 2) there are more (less) particles with spherical nature alongside lifted rain drops in the observations that reduce the total $Z_{DR}$ compared to Thompson/P3 (Morrison/FSBM). Furthermore, 3) the observed variability of $Z_{DR}$ is possibly not correctly captured by the radar forward simulator which has to assume fixed distributions of particle orientations as well as a fixed aspect ratio of the particles.

At upper levels clear differences between Morrison/FSBM and Thompson/P3 can be seen. Morrison and the FSBM scheme
show $Z_{DR}$ values of up to 4 dB at these heights while the Thompsons and the P3 schemes are close to 0 dB. Here, the high $Z_{DR}$ are caused by cloud ice (see Appendix B for CFADs of radar signals separated by hydrometeor class). All schemes assuming spherical cloud ice or with other dominating spherical hydrometeor classes at these heights show small $Z_{DR}$. This is true for the P3 small ice fraction for which the forward simulator assumes spherical aspect ratio of 1. In the Thompson schemes, the assumed aspect ratio by the forward simulator is 0.2, suggesting that other hydrometeor classes with lower $Z_{DR}$ like snow
or graupel dominate the signal. Only for FSBM and Morrison (aspect ratio 0.2) cloud ice dominates the signal. The stronger signal in FSBM and Morrison is not a result of different density assumptions, because both, the FSBM and Morrison scheme assume lower density of cloud ice compared to Thompson. The observations do not show increased $Z_{DR}$ at these heights. This could either mean that 1) there are no large cloud ice particles observed, 2) that the signal is dominated by other more spherical particles in the observations, or 3) that the assumed aspect ratio of 0.2 by the radar forward operator is unrealistic and the
observed particles are more spherical in nature.

### 3.4 Profiles of Dual-Wavelength variables

More insight about the particle size is provided by the simulated and observed dual-wavelength ratio DWR. The standard radar reflectivity is strongly influenced by the number of particles within the radar beam, the particles sizes and the particles densities. In contrast, DWR is rather sensitive to the particles size. In principle, it is also sensitive to the particle density, but
the simulated density is assumed to be constant or a function of particle size. Figure 7 shows deviations between the schemes within the ice phase as well as in the liquid phase. Here, no attenuation correction is applied. This makes the comparison to the radar observations less realistic but reveals differences in microphysical processes and fingerprints between the simulations more clearly. The observations show DWR close to 0 at upper levels where ice crystals are very small. All simulations and the observations agree at these heights. The observations then show a steady increase of DWR towards the melting layer height.
This is reflecting ice particle growth, given that DWR is mainly sensitive to particle size. All simulations reproduce this increase of DWR towards the melting layer heights but differ in the slope and height where the increase starts. While the Morrison and FSBM simulations show a beginning increase in DWR already at about 10 km, the P3 and the two Thompson simulations show

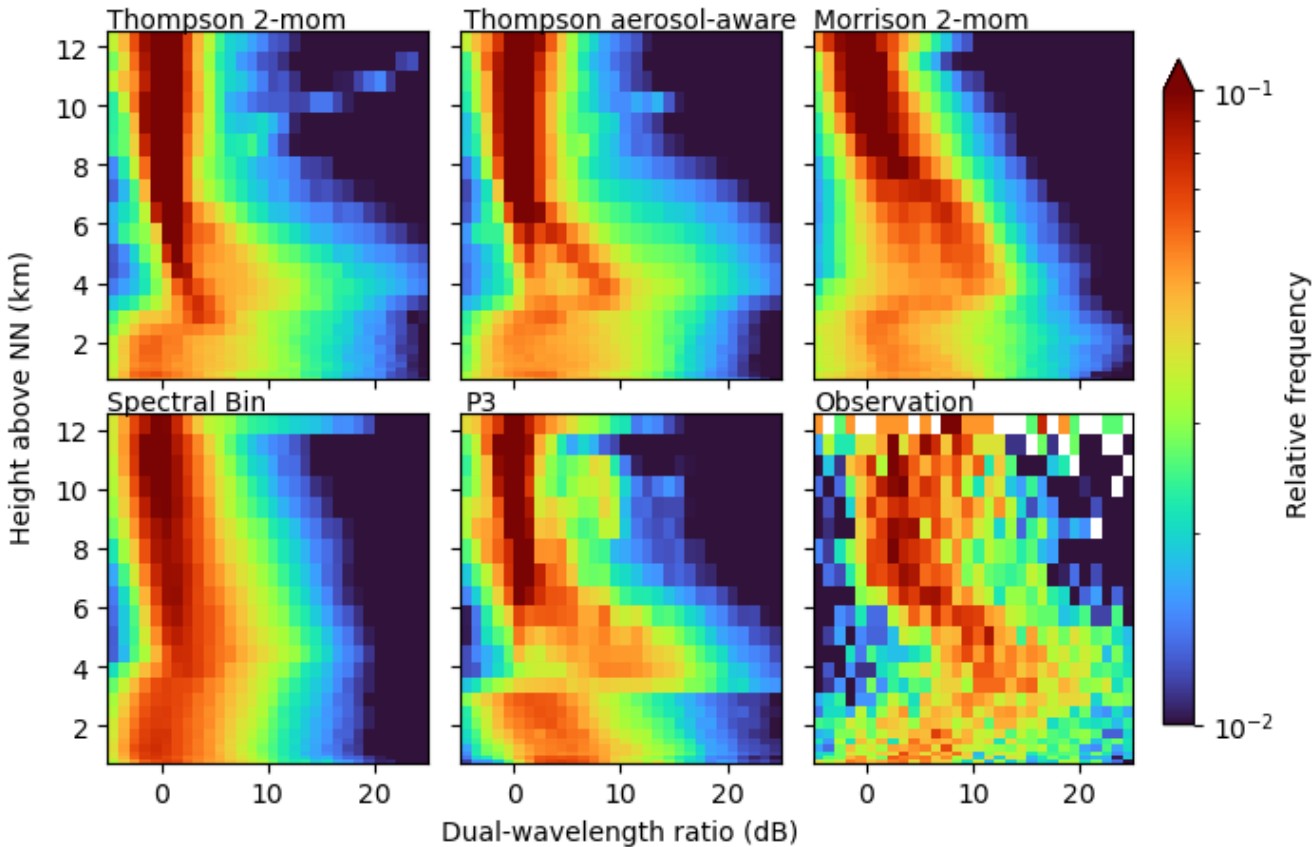

**Figure 7.** CFADs of simulated and measured DWR over 5 convective days in 2019. Radar observations with Poldirad and Mira35 (Poldirad - Mira35).

a beginning increase at about 7 km, which better agrees with the observations. At melting layer heights the DWR values reach their maximum in all simulations. The magnitude of the maximum DWR values differs: Morrison and FSBM do not produce DWR larger than 20 dB, while the P3 and the two Thompson schemes produce DWR of up to 25 dB. At these heights the two Thompson schemes produce distinct streaks of higher frequencies at low DWR values (0-10 dB) and then a diffuse area of lower frequencies at higher DWR (> 10 dB). The streaks are related to snow growth during sedimentation. Thompson only uses one mass-size relation for snow while the P3 is more flexible: it uses a varying mass-size relation depending on whether the ice particle is unrimed, partially rimed or fully rimed. This is the reason why the DWR corridor in P3 above the melting layer height is wider.

Below the melting layer the observed DWR steadily decreases towards the ground. The models do not reproduce this very well: Even though the DWR decreases in all models, this decrease happens abruptly at the melting layer. The DWR directly below the melting layer height is very different between the models and the observations. However, including attenuation in-

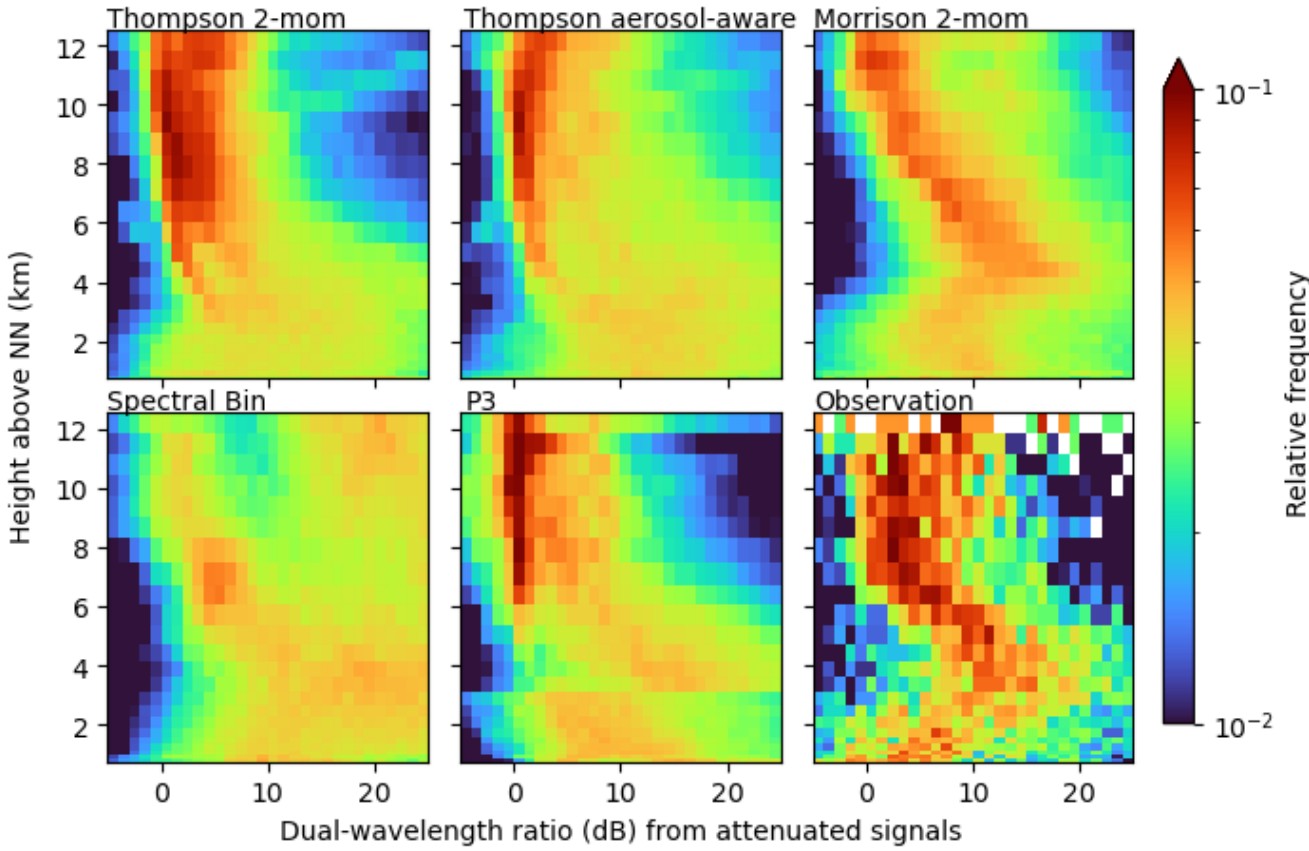

**Figure 8.** Same as Fig. 7 but with simulated attenuation included in the simulations.

creases the simulated DWR and its variability, making it difficult to quantify DWR deviations between model and observations
as discussed below. Below the melting layer height the simulated DWR stays more or less constant while the observed decreases towards the surface. In the P3 simulations (and weaker in the Morrison scheme) the DWR even increases again towards the ground. At these heights rain and graupel are the dominant species. The simulated increase of DWR towards the ground is likely a result of the simulated collection process: Rain droplets grow while falling by collecting smaller droplets. This is visible also directly in the rain PSD (see appendix B) and was discussed in the previous section 3.3. Opposed to this, the large
particles precipitating from the melting layer seem to shrink towards the ground, perhaps by drop breakup or evaporation. The general magnitude of simulated DWR near the surface is close to the observed again at around -3 to 10 dB.

Comparing DWR signatures without including attenuation is giving insight into the details of the microphysical schemes but is not well suited for a direct comparison with radar observations, because especially the Ka-band observations are potentially strongly attenuated. This would lead to an increase in DWR. Figure 8 shows the same DWR CFADs including attenuation.
Obviously attenuation drastically increases the variability in DWR. As a result, DWR values are scattered over larger ranges,

partially masking the underlying fingerprints that were visible in the CFADs without attenuation. Furthermore, the lower number of observed profiles compared to the simulated profiles is still clearly visible. This is most prominent in the DWR CFAD but also in the CFADs of reflectivity (Fig. 5) and $Z_{DR}$ (Fig. 6). This is reminiscent of the large observational effort to collect targeted cell core RHI scans.

## 4   Summary and conclusions

A methodological framework has been presented that allows for a statistical comparison of polarimetric dual-wavelength radar observations with numerical weather model output. Targeted dual-radar observations of convective cell characteristics in the vicinity of Munich over a significant fraction of cell lifetime have been established. For the weather model, cell specific observations of a Ka-band and two C-band radars are produced using a polarimetric radar forward operator and automatic
cell-tracking. The total data set presented includes 30 convective days of simulation and radar observations so far. Targeted dual-wavelength observations were performed on 5 of those days, adding up to about 1300 RHI profiles of dual-wavelength observations of convective clouds. A convection permitting WRF setup over Munich has been implemented. WRF hindcast simulations were conducted with 5 different microphysics schemes of varying complexity. The radar forward simulator CR-SIM was applied that provides polarimetric radar variables from model output. The cell-tracking algorithm TINT was applied
on radar and model data in the same way to allow for comparison of convective cell characteristics.

With the exception of the Thompson schemes, all microphysics schemes simulate too few convective cells compared to the radar observations. The difference is mainly caused by missing weak cells with cell core heights between 3-4 km. This points to missing small scale development in most of the simulations. This suggests dynamical reasons or numerical resolution issues rather than impact of microphysics schemes. It is a reminder that the presented methodology is not able to perfectly
separate microphysics from dynamical feedbacks or other impacts. Statistics of observed and simulated cell reflectivities show that models are not able to reproduce the observed high occurrences of very weak cells, as well as the occurrence of strongest reflectivities at more than 57 dBZ. This might be related to numerical smoothing of local and rare values in the NWP model.

Targeted scans of convective clouds revealed differences in radar observations and model as well as between microphysics schemes in ice and liquid phases. Overall the schemes agree in simulated reflectivity in below the melting layer height, but
comparison to the radar observations show that reflectivity in this area is overestimated in all schemes. Simulated $Z_{DR}$ reveal that only the FSBM scheme is able to reproduce the radar signatures reasonably well, all other schemes produce $Z_{DR}$ signals within rain with too much spread. This is likely a result of the assumed exponential (Marshall-Palmer) rain PSD producing too many large and too few small droplets. The FSBM scheme on the other hand demonstrates the advantage of explicitly resolving the PSD that results in a more realistic radar signature in rain.
Above the melting layer height more deviations between the schemes are found. The P3 scheme is the only scheme not overestimating reflectivities directly above the melting layer height. All other schemes show unrealistic high reflectivities related to graupel, partly over 45 dBZ. This means that either too many or too large graupel particles are produced or that the density assumption of graupel is set too high by the Morrison, FSBM and especially the Thompson schemes. An overestimation

of reflectivities in the ice phase was found by other studies as well who attribute this bias to wrong graupel and snow particle size distributions (e.g., Stanford et al., 2017; Varble et al., 2011). CFADs of DWR allow to analyze the simulated particles sizes more directly. The two Thompson schemes produce clearly confined distributions of higher occurrences of DWR in the ice phase related to snow and its growth by aggregation during sedimentation. The P3 scheme also produces distinct narrow distribution of DWR values at these heights. Nonetheless it is wider compared to the Thompson results, caused by a mixture of unrimed, partly rimed and unrimed particles. We believe this demonstrates the greater flexibility of the P3 scheme in the ice phase where this scheme deviates from the traditional hydrometeor classes and instead predicts properties for a single ice category. This seems to produce a wider range of particle characteristics (and hence DWR signals) as opposed to the other schemes, where most of the signal is produced along distinctly visible corridors. CFADs of $Z_{DR}$ reveal deviations at the highest levels above 10 km, where the Morrison and FSBM scheme produce larger values related to cloud ice that are not visible in the observed radar signal. This could either be a result of simulated cloud ice particles being too large or too many, but this could also be a result of the assumed flat cloud ice shape with an aspect ratio of 0.2. Directly above the melting layer height, three schemes (the two Thompsons and the P3) show increased $Z_{DR}$ signals that are dominated by large lifted rain drops, while Morrison and FSBM do not show any increase of $Z_{DR}$. The observations are between both extremes. This suggests that there are fewer large drops in the observation than simulated by Thompson/P3 or that their signals are not dominated by lifted rain drops but other spherical particles reduce the total measured $Z_{DR}$.

In general, we could demonstrate how weather simulations with varying microphysics schemes produce varying polarimetric and DWR radar signatures. However, one interesting fact is that the two Thompson schemes do not show significant differences from one another. Even though the schemes are very similar, one could have expected that the explicit prediction of droplet number concentration as well of aerosol variables would have a stronger influence on the weather simulation.

Using our framework, there are some challenges for the evaluation of the microphysics schemes performance. Using a large data set provides the possibility of a statistical evaluation. Thus, it can provide correct general overview of the schemes performance. On the other hand, considering long periods of time, multiple different weather situations produce convective cells of varying types. In our analysis, these are all analyzed together. This introduces ambiguities and some individual microphysical aspects might be smeared out. A solution would be a separation of different convective cloud types, e.g. by classifying into shallow, congestus or deep convective clouds using our 32 dBZ echo top height (e.g., Matsui et al., 2009). Furthermore, classifications into weak/strong forcing situations could be of interest, to analyze the effect of, e.g., frontal systems on the distribution of radar signals. This will be addressed in a future application of this framework.

Furthermore, there are uncertainties connected to the radar forward simulator applied. To calculate scattering characteristics, assumptions have to be made including the particles aspect ratio, orientations, shape and more. The variability of the simulated signals is reduced by applying fixed relations compared to the potential variability of shapes, orientations and aspect ratios in nature. In addition, the radar forward simulator applied in our study does not consider mixed phase particles. This means that, e.g., effects such as the bright band where particles melt cannot be reproduced by the simulations. To circumvent some ambiguities introduced this way, the comparison could be extended from radar signal space to cloud hydrometeor space. I.e., retrieved hydrometeor classes can be compared to simulated ones.

Finally, there is more noise in our radar statistics compared to the simulation statistics (for example Figure 5) due to the lower number of data points available from the observations. This could partially explain biases between model and radar, reminiscent of the large observational effort to statistically compare convective cloud characteristics.

The analyses shown in this work demonstrate the potential to analyze the treatment of small scale processes within microphysics schemes. More analyses will be conducted with the methods presented, especially including dual-wavelength and polarimetric variables to analyze the simulated particle shapes and sizes. The observed radar CFADs still show large scatter due to small numbers of measurements included. More dual-wavelength data is needed to compare radar observations and a model for convective weather situations with more confidence. Another operational dual-wavelength measurement strategy is currently being established that makes use of the operational DWD volume scans and copies their strategy with the Mira-35 Ka-band radar. Because the volume scan strategy consists of multiple PPI scans of different elevations angles, the vertical resolution will be somewhat lower compared to our dual-wavelength RHI scans in strategy B. On the other hand, the PPI strategy possibly samples multiple cells at the same time and together with the operational setup we expect to obtain a larger number of dual-wavelength measurements of convective cells.

Based on the methodology presented in this paper, more detailed analysis of some of the observed differences will be analyzed next. This will allow to slowly approach the answer to the question which level of complexity in microphysical processes needs to be implemented to realistically represent cloud and precipitation distribution in NWP models at the same time.

*Code and data availability.* The polarimetric radar data from the operational C-band radar in Isen is available for research from the German Weather Service (DWD) upon request. The Poldirad and Mira-35 data presented in this paper is available through the authors upon request. Data of WRF, CR-SIM, and TINT simulations is also available through the authors upon request. The software developed for this paper is available here: doi.org/10.5281/zenodo.5526882. The Weather Research and Forecasting Model (WRF; version 4.2) is openly available on github: https://github.com/wrf-model/WRF. The cell-tracking algorithm TINT is openly available on github: https://github.com/openradar/TINT (last accessed 21.09.2021). The forward operator CR-SIM (version 3.33) is available at the website of the Stonybrook University (https://you.stonybrook.edu/radar/research/radar-simulators/; last accessed 21.09.2021).

# Appendix A: Simulation and Observation dates

**Table A1.** List of convective days that were used in our analyses. Strategy A always refers to the whole day.

| Date | Strategy |
|------|----------|
| 29.04.2019 | Strategy A |
| 06.05.2019 | Strategy A |
| 28.05.2019 | Strategy A, Strategy B (11:25 - 14:00 UTC) |
| 29.05.2019 | Strategy A |
| 11.06.2019 | Strategy A |
| 12.06.2019 | Strategy A |
| 21.06.2019 | Strategy A, Strategy B (14:40 - 17:25 UTC) |
| 01.07.2019 | Strategy A, Strategy B (11:20 - 16:50 UTC) |
| 07.07.2019 | Strategy A, Strategy B (09:20 - 15:10 UTC) |
| 08.07.2019 | Strategy A, Strategy B (09:00 - 14:00 UTC) |
| 17.06.2020 | Strategy A |
| 20.06.2020 | Strategy A |
| 27.06.2020 | Strategy A |
| 28.06.2020 | Strategy A |
| 29.06.2020 | Strategy A |
| 01.07.2020 | Strategy A |
| 10.07.2020 | Strategy A |
| 11.07.2020 | Strategy A |
| 23.07.2020 | Strategy A |
| 24.07.2020 | Strategy A |
| 26.07.2020 | Strategy A |
| 28.07.2020 | Strategy A |
| 01.08.2020 | Strategy A |
| 02.08.2020 | Strategy A |
| 03.08.2020 | Strategy A |
| 18.08.2020 | Strategy A |
| 17.09.2020 | Strategy A |
| 22.09.2020 | Strategy A |
| 23.09.2020 | Strategy A |
| 12.10.2020 | Strategy A |

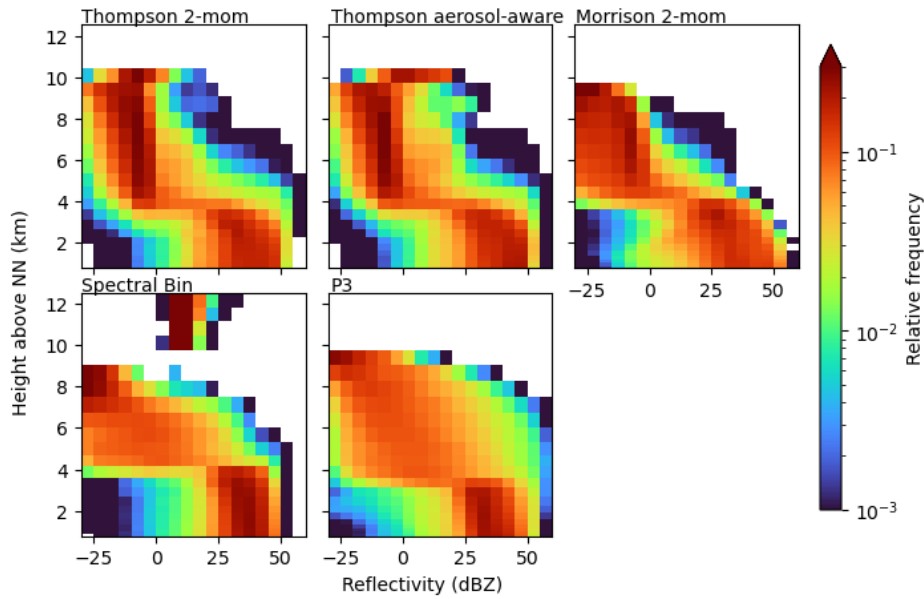

**Figure B1.** CFADs of simulated reflectivity of the rain hydrometeor class over 5 convective days in 2019.

## Appendix B:  Hydromeoteor class CFADs

CR-SIM calculates radar signals for the single hydrometeor classes independently, next to the total signal of all hydrometeors together. Below are CFADs of the signals calculated from the most relevant hydrometeor classes. The CFADs are shown on the original WRF grid and without attenuation correction. The FSBM simulation sometimes showed spurious rain signals on the highest levels (> 10 km). Sometimes there are small numbers of rain drops present in the largest bins, even though the mass mixing ratio of rain is 0 in the FSBM simulation. We consider this an error with no physical meaning.

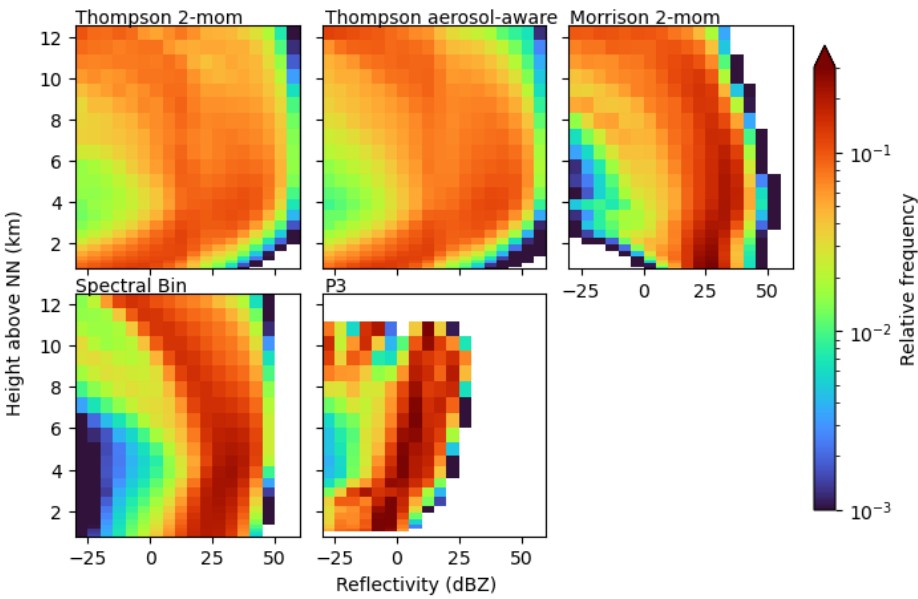

**Figure B2.** CFADs of simulated reflectivity of the graupel hydrometeor class over 5 convective days in 2019.

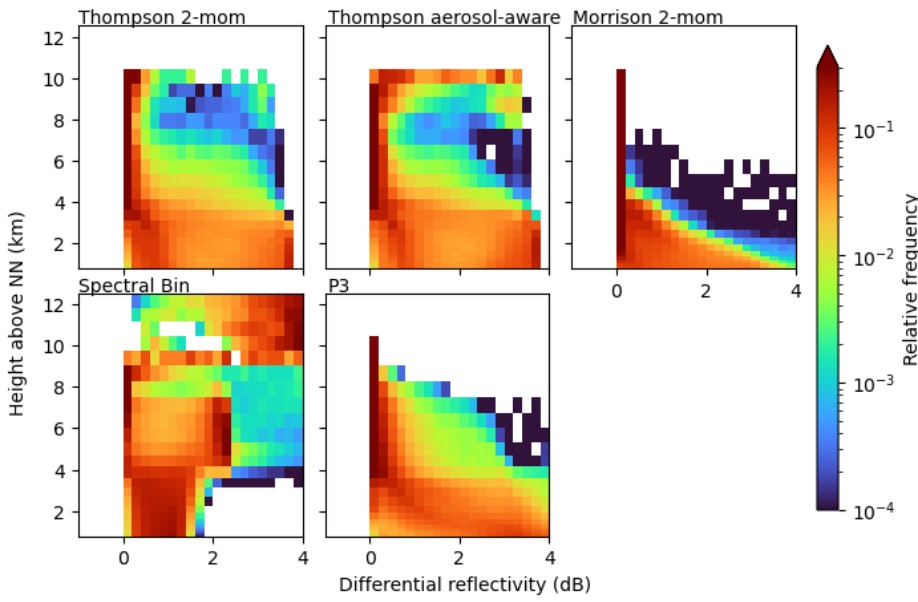

**Figure B3.** CFADs of simulated differential reflectivity of the rain hydrometeor class over 5 convective days in 2019.

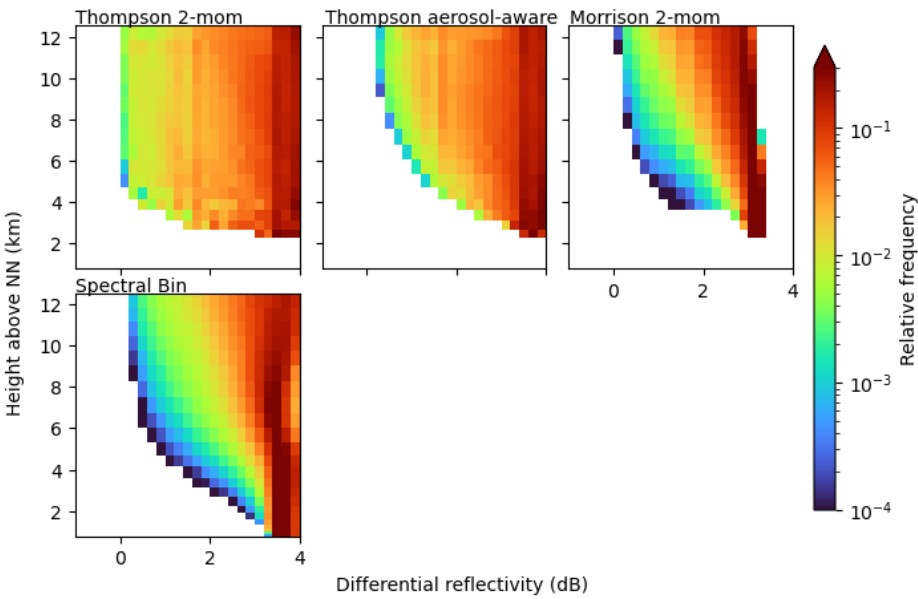

**Figure B4.** CFADs of simulated differential reflectivity of the cloud ice hydrometeor class over 5 convective days in 2019. The P3 scheme does not provide the classical cloud ice category.

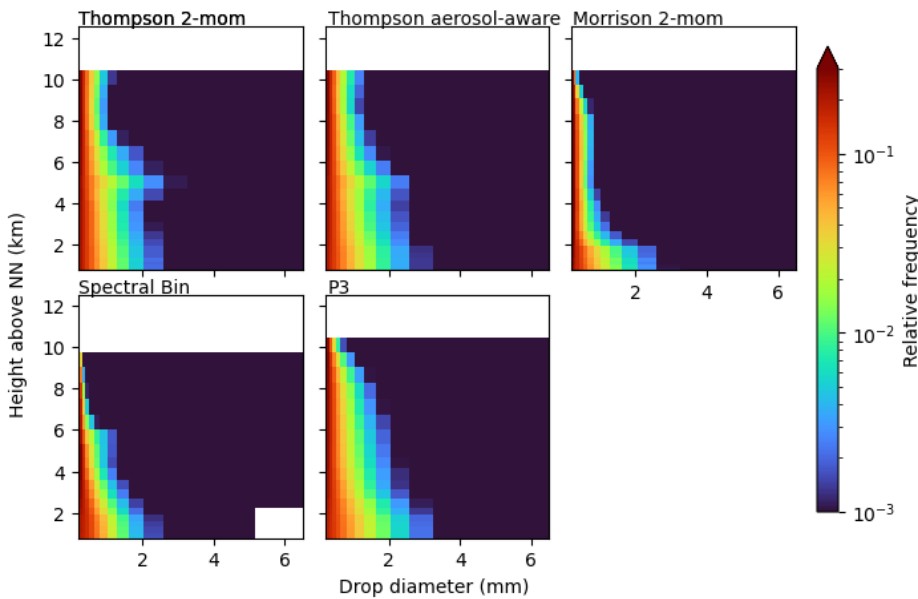

**Figure B5.** CFADs of simulated rain drop size distributions over during the measurement periods over 5 convective days in 2019.

*Author contributions.* Eleni Tetoni, Florian Ewald, Martin Hagen and Gregor Köcher performed radar measurements during precipitation events. Gregor Köcher developed the methodology presented and wrote the manuscript in its current form. Tobias Zinner and Christoph Knote supervised and discussed the scientific content. All authors commented on the manuscript.

*Competing interests.* The authors declare that they have no conflict of interest.

*Acknowledgements.* We gratefully acknowledge the project "Investigation of the initiation of convection and the evolution of precipitation
using simulations and polarimetric radar observations at C- and Ka-band" (IcePolCKa; Grant ZI 1132/5-1 and HA 3314/9-1) project funded by the German Research Foundation (DFG) as part of the special priority program on the Fusion of Radar Polarimetry and Atmospheric Modelling (SPP-2115, PROM). Mariko Oue and Aleksandra Tatarevic were a great support for the implementation and adjustment of CR-SIM. We want to thank Fabian Hoffmann and Bernhard Mayer for their comments on the manuscript. We would also like to thank Toshi Matsui and one anonymous reviewer for their comments that improved the quality of the manuscript.

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
