# Peer review of "Evaluation of convective cloud microphysics in numerical weather prediction model with dual-wavelength polarimetric radar observations: methods and examples"

_Atmospheric Measurement Techniques, 2021_

## Author Comment (AC1)

**Title**: Evaluation of convective cloud microphysics in numerical weather prediction model with dual-wavelength polarimetric radar observations: methods and examples

**Author**(s): Gregor Köcher, Tobias Zinner, Christoph Knote, Eleni Tetoni, Florian Ewald, and Martin Hagen

**Summary**

This study examines a series of regional storm-resolving weather forecasting with different microphysics schemes against long-term observations of dual-wavelength polarimetric radars over Munchen area. Although there are many papers inter-compare microphysics schemes available in WRF, the noble aspect of this study is to utilize polarimetric radar simulator and cell-tracking algorithm for more consistent sampling of polarimetric radar observables and dual-wavelength ratio. However, there are some major questions/suggestions related to 1) missing citations, 2) separation of convective cells, and 3) actual rain drop-size distributions, and 4) uncertainties of the forward model. By improving these issues, this manuscript could be quite powerful. Thus, my suggestion is "major revisions" in order to publish in ACP.

We like to thank the reviewer for the valuable comments which helped to improve the manuscript quality substantially. Please find below our point-by-point reply highlighted in blue. A marked-up manuscript version showing the changes made is provided along with the revised manuscript.

**Major Comments/Suggestions**

1) Missing citations

This study cites several microphysics evaluation papers, but it completely misses previous studies directly using forward polarimetric radar models. Here are suggested references. Please take a look and relate yours to their findings and approaches.

Jung, Y., M. Xue, and G. Zhang, 2010: Simulations of Polarimetric Radar Signatures of a Supercell Storm Using a Two-Moment Bulk Microphysics Scheme. *J. Appl. Meteor. Climatol.,* **49**, 146–163, https://doi.org/10.1175/2009JAMC2178.1

Snyder, J.C., H.B. Bluestein, D.T. Dawson II, and Y. Jung, 2017a: Simulations of Polarimetric, X-Band Radar Signatures in Supercells. Part I: Description of Experiment and Simulated ρhv Rings. *J. Appl. Meteor. Climatol.,* **56**, 1977–1999, https://doi.org/10.1175/JAMC-D-16-0138.1

Ryzhkov, A., M. Pinsky, A. Pokrovsky, and A. Khain, 2011: Polarimetric Radar Observation Operator for a Cloud Model with Spectral Microphysics. J. Appl. Meteor. Climatol., 50, 873–894. doi: http://dx.doi.org/10.1175/2010JAMC2363.1

Putnam, B.J., M. Xue, Y. Jung, G. Zhang, and F. Kong, 2017: Simulation of Polarimetric Radar Variables from 2013 CAPS Spring Experiment Storm-Scale Ensemble Forecasts and Evaluation of Microphysics Schemes. Mon. Wea. Rev., 145, 49–73, https://doi.org/10.1175/MWR-D-15-0415.1

Also a following citation is recommended for microphysics finger print from polarimetric radar signals. This Ph.D. dissertation describes various microphysics finger prints related to cloud microphysics processes.

Kumjian, M.R., 2012: The impact of precipitation physical processes on the polarimetric radar variables. PhD. Dissertation, University of Oklahoma, 327 pp.

Thank you for your suggestions. We included the proposed literature regarding polarimetric radar forward operators to evaluate cloud model microphysics in our introduction:

> *There are some studies that directly use polarimetric radar forward operators to evaluate the performance of cloud microphysics schemes. For instance, Jung et al. (2010) and Snyder et al. (2017) each simulate idealized super cell events to test if the cloud microphysics schemes together with a polarimetric radar forward operator are able to reproduce known super cell radar signatures. Ryzhkov et al. (2011) and Putnam et al. (2017) compare simulated polarimetric radar signals with radar observations to evaluate microphysics schemes, but focus on one or two convective cases.*

Furthermore, we have related the findings of Ryzhkov et al. (2011) and Putnam et al. (2017) to ours in the discussion. Jung et al. (2010) and Snyder et al. (2017) simulate idealized super cell events, which is a different approach than ours. That's why we think it is enough to relate our approach to theirs in the introduction, but we don't relate our findings to theirs in the discussion.

Putnam et al. (2017), section 3.2:
> *This is in agreement with Putnam et al. (2017) who compare radar signals simulated by 5 different microphysic schemes for two case studies and find that especially the Morrison scheme but to a lesser extent also the Thompson scheme produces too high Z. They attribute this to stratiform rain PSDs that contain too many large drops, to an overforecast of the precipitation coverage overall and in case of Morrison, to a high bias of wet graupel in convective regions. Given that the forward simulator applied in this study does not consider wet particles, we find the high bias in Z exists even without considering wet graupel and comes mostly from rain, suggesting PSDs that contain too many large rain drops compared to the observations.*

Putnam et al. (2017), section 3.3:
> *Putnam et al. (2017) find similar results regarding ZDR signatures near the surface: in their two case studies, the simulations with Thompson and with Morrison cloud microphysics showed incorrect ZDR maxima associated with isolated large drops at locations of weak convection where this would not be expected.*

Ryzhkov et al. (2011), section 3.3:
> *Ryzhkov et al. (2011) for example evaluate radar signals simulated from a spectral bin scheme against a hailstorm case and find that their spectral bin scheme produces PSDs for rain that deviate from the gamma distribution. Bulk schemes would not be able to reproduce these PSDs and since radar signals strongly depend on the PSD, Ryzhkov et al. (2011) argue that spectral bin schemes are better suited to simulate polarimetric radar signals.*

Finally, we referenced the Ph.D. Dissertation about fingerprints in polarimetric radar signals, also in our introduction. This Ph.D. Dissertation helps a lot in understanding the impact of rain processes on polarimetric radar signals:

> *Kumjian (2012) demonstrate the impact of precipitation processes on polarimetric radar signals, though he focuses mainly on rain processes, such as raindrop evaporation or size sorting.*

2) separation of convective cells

One of the advantages of this paper is the large sampling volume, but it simultaneously induces ambiguity for analysis. During a long-term period, there must be various sizes of convections from shallow, congestus, and deep convective/stratiform cells. When you bundle all into a single CFAD, it tends to smear out important aspects of microphysics. Please check the following papers on how it separates cloud type and better evaluates different aspects of microphysics from long-term simulations/observations.

Matsui, T., X. Zeng, W.-K. Tao, H. Masunaga, W. Olson, and S. Lang (2009), Evaluation of long-term cloud-resolving model simulations using satellite radiance observations and multifrequency satellite simulators. *Journal of Atmospheric and Oceanic Technology*, 26, 1261-1274.

You can use echo-top height from each cell-tracked target for separation. But if this type of separation is too difficult to implement (or too much effort), please just discuss and try it in the future.

We have added a part discussing this topic in the conclusions. This is a valuable comment, we plan to include a separation of convection type for the next steps of analysis of the data to be published soon after this manuscript. Apart from a possible separation using the echo-top height for different convection sizes, we think a separation into weak forcing / strong forcing situations could be interesting. We think that this is too much effort for the present work, that's why we are just discussing it in this manuscript and will add it for future publications.

Section 4:

> *Using our framework, there are some challenges for the evaluation of the microphysics schemes performance. Using a large data set provides the possibility of a statistical evaluation. Thus, it can provide correct general overview of the schemes performance. On the other hand, considering long periods of time, multiple different weather situations produce convective cells of varying types. In our analysis, these are all analyzed together. This introduces ambiguities and some individual microphysical aspects might be smeared out. A solution would be a separation of different convective cloud types, e.g. by classifying into shallow, congestus or deep convective clouds using our 32 dBZ echo top height (e.g., Matsui et al., 2009). Furthermore, classifications into weak/strong forcing situations could be of interest, to analyze the effect of, e.g., frontal systems on the distribution of radar signals. This will be addressed in a future application of this framework.*

3) rain drop-size distributions

Probably the most robust finding in this study is the variability of rain-DSD related radar signals (ZDR and DWR) among different microphysics schemes as seen in Figures 5, 6, and 7. Above-melting-zone evaluations tend to have more uncertainties in the forward model (described in next). To augment your finding in radar signals and discussion, it's much better to directly examine simulated rain drop size distribution profiles (like CFAD format) from different microphysics schemes. This should not be a difficult task. ( it's much better if you have disdrometer observations!)

We provided a CFAD of rain drop size distributions for convective cells (Appendix B):

[Figure]

The following passage was added to section 3.3:

> *In order to separate the analysis into reasons due to differences in the underlying modeled microphysics and due to different processing in the forward simulator, we examined rain particle size distributions directly produced by the NWP model (Rain PSD CFAD in Appendix B.) The FSBM scheme provides  the drop size distributions over a number of size bins, for the other schemes we calculated the distributions according to the schemes parameterization. Only model grid boxes that were flagged as a convective cell by the TINT cell tracking are considered. The rain PSD CFAD confirms the findings of the ZDR CFAD: the two Thompson schemes simulate large rain drops from the surface up to the melting layer height and even above, while the Morrison scheme produces large rain drops only at the surface and the FSBM produce the highest frequency of small drops.*

4) uncertainties of forward model

Details and uncertainties in assumptions of the forward model (CR-SIM) are not discussed. In order to represent simulated microphysics in polarimetric observables, one must assume particle shape and orientation simultaneously in the forward model, because these are not "explicitly" simulated in most of the microphysics schemes in WRF. Following paper discusses and tests different

assumptions. Please describe what kind of shape/orientation assumptions are made for each microphysics in Section 2.4 (or Appendix), and related discussions in Section 3.3.

Matsui, T., Dolan, B., Rutledge, S. A., Tao, W. â ⚲ ⚳ K., Iguchi, T., Barnum, J., & Lang, S. E. (2019). POLARRIS: A POLArimetric Radar Retrieval and Instrument Simulator. *Journal of Geophysical Research: Atmospheres*, 124. https://doi.org/10.1029/2018JD028317

We have not been clear on the assumptions of the radar forward model (CR-SIM). The section of CR-SIM (2.4) has been appended with the assumptions that CR-SIM is making regarding particle shapes, particle orientation and dielectric constants for each microphysics schemes:

> *The dielectric constant of water is 0.92. Solid phase hydrometeors are assumed to be dielectric dry oblate spheriods and are represented as a mixture of air and solid ice. The refractive index hence depends on the hydrometeor density and is computed using the Maxwell-Garnet (1904) mixing formula. There are no mixed phased particles simulated. This means mixed phase radar signatures (for example the "bright band", Austin and Bemis, 1950) will not be reproduced by the simulation. In order to simulate polarimetric radar observables, a radar forward simulator must assume particle shapes and particle orientation. The particle orientation assumptions are the same for all schemes. It is assumed that the particle orientations are 2D Gaussian distributed with zero mean canting angle as in Ryzhkov et al. (2011). The width of the angle distributions is specified for each hydrometeor class: 10° for cloud, rain, and ice and 40° for snow, unrimed ice, partially rimed ice, and graupel. Regarding the shape assumptions, cloud droplets are simulated as spherical (aspect ratio of 1) and raindrops are simulated as oblate spheriods with a changing axis ratio dependent on the drop size according to Brandes et al. (2002) in all schemes. For ice hydrometeor classes, the same aspect ratio assumptions are applied for all schemes except the P3 scheme: cloud ice is assumed as oblate with a fixed aspect ratio of 0.2. Snow is assumed as oblate with a fixed aspect ratio of 0.6. Graupel is assumed to be oblate with an aspect ratio that is changing from 0.8 to 1, dependent on the diameter and according to Ryzhkov et al. (2011):*

> $ar = 1.0 - 0.02$      *if D < 10 mm ,*
> $ar = 0.8$          *if D > 10 mm .*

> *The P3 scheme does not provide the standard ice hydrometeor classes. Instead, the aspect ratio of small ice (spherical, fixed aspect ratio of 1), unrimed ice (oblate, fixed aspect ratio of 0.6), partially rimed ice (oblate, fixed aspect ratio of 0.6) and graupel (spherical, fixed aspect ratio of 1) is assumed by CR-SIM. This means in comparison to the other schemes that the P3 simulation deviates for small ice (aspect ratio of 1 in P3, while cloud ice in other schemes is assumed to have an aspect ratio of 0.2) and graupel (0.8 - 1 in other schemes, while graupel particles in P3 are assumed to have an aspect ratio of 1). Resulting differences in the radar signal are discussed in the result section 3 whenever it might influence the simulated radar signal.*

We further added a small abstract in the conclusions acknowledging uncertainties produced by the radar forward operator:

> *Furthermore, there are uncertainties connected to the radar forward simulator applied. To calculate scattering characteristics, assumptions have to be made including the particles aspect ratio, orientations, shape and more. The variability of the simulated signals is reduced by applying fixed relations compared to the potential variability of shapes,*

*orientations and aspect ratios in nature. In addition, the radar forward simulator applied in our study does not consider mixed phase particles. This means that, e.g., effects such as the bright band where particles melt cannot be reproduced by the simulations. To circumvent some ambiguities introduced this way, the comparison could be extended from radar signal space to cloud hydrometeor space. I.e., retrieved hydrometeor classes can be compared to simulated ones.*

Finally, we added multiple parts in the discussion and conclusions that relate forward simulator details to the results:

Section 3.2

*Given that the forward simulator applied in this study does not consider wet particles, we find the high bias in Z exists even without considering wet graupel and comes mostly from rain, suggesting PSDs that contain too many large rain drops compared to the observations.*

Section 3.3:

*Furthermore, 3) the observed variability of ZDR is possibly not correctly captured by the radar forward simulator which has to assume fixed distributions of particle orientations as well as a fixed aspect ratio of the particles.*

Section 3.3:

*All schemes assuming spherical cloud ice or with other dominating spherical hydrometeor classes at these heights show small ZDR. This is true for the P3 small ice fraction for which the forward simulator assumes spherical aspect ratio of 1. In the Thompson schemes, the assumed aspect ratio by the forward simulator is 0.2, suggesting that other hydrometeor classes with lower ZDR like snow or graupel dominate the signal. Only for FSBM and Morrison (aspect ratio 0.2) cloud ice dominates the signal. The stronger signal in FSBM and Morrison is not a result of different density assumptions, because both, the FSBM and Morrison scheme assume lower density of cloud ice compared to Thompson. The observations do not show increased ZDR at these heights. This could either mean that 1) there are no large cloud ice particles observed, 2) that the signal is dominated by other more spherical particles in the observations, or 3) that the assumed aspect ratio of 0.2 by the radar forward operator is unrealistic and the observed particles are more spherical in nature.*

Section 4:

*This could either be a result of simulated cloud ice particles being too large or too many, but this could also be a result of the assumed flat cloud ice shape with an aspect ratio of 0.2.*

**Minor Comments/Suggestions**
Line 25: "the huge number" -> "a large number"

Changed as suggested.

Line 26: "on scales of μm to mm and" -> "on scales of mm or smaller" for consistency. In fact, microphysics processes occur less than the scale of micron, such as ice crystallization processes.

Changed as suggested.

Lin 82: "with a sound statistical basis" ?? I don't understand.

By sound statistical basis we mean a large sample size. We changed the phrase for clarification:

> *2. Evaluate multiple state-of-the-art cloud microphysics schemes for current generation numerical weather prediction models in a common model framework against observations with a large sample size.*

Line 89: "separate the microphysical impacts from possible feedbacks." I agree. But more bottom line, I would argue whether your set of numerical weather model resolved dynamics or not with 2km horizontal grid spacinig.

There is a misunderstanding here. The middle domain of our model setup has a 2 km horizontal grid spacing. The inner domain that we used for the analysis has a grid spacing of 400 m. We assume you refer to the cloud dynamics which we believe are resolved at a grid spacing of 400 m. That's why we left this sentence as it is. Perhaps the comparison to current operational weather models (line 150) led to the confusion of our horizontal grid spacing. We added a sentence clarifying that our grid spacing is better than current operational numerical weather prediction models and is rather representing the future generation of NWP models in section 2.2:

> *Currently, operational limited area weather models operate at 2 km grid spacing (e.g., 2.8 km in COSMO-DE of the German Weather Service; Baldauf et al., 2011) which means our inner domain has a resolution that is effectively about 5 times higher and should be representing the future generation of operational limited area weather models.*

Line 108: "frequency" -> "frequencies"

Changed as suggested.

Line 149 and repeat the same: "a horizontal resolution of 10 km," must be replaced by "horizontal grid spacing of 10km". Numerical atmospheric model does not have 10km resolution with 10km of horizontal grid spacing. Effective dynamic resolutions are x5 ~ x10 of horizontal grid spacing in numerical dynamic core. Apply this correction elsewhere in the manuscript.

Pielke, R. A. (1991). A Recommended Specific Definition of "Resolution", Bulletin of the American Meteorological Society, 72(12), 1914-1914. Retrieved Oct 30, 2021, from https://journals.ametsoc.org/view/journals/bams/72/12/1520-0477-72_12_1914.xml

Thank you for this input. Even though we think that 'resolution' is a widely accepted term instead of grid spacing, we understand the logic behind your comment and adapted the definition of the reference throughout the manuscript concerning the model grid descriptions.

Line 159: Please briefly describe other physics options, such as land surface, PBL, and radiation schemes.

The namelist of WRF is added as a supplement where all options used can be seen. The requested information about land surface, PBL and radiation scheme has been added to section 2.2:

> *Other physics options include the Noah Land Surface model (Ek et al., 2003; Chen and Dudhia, 2001), the MYNN2 planetary boundary layer scheme (Mellor-Yamada scheme by Nakanishi and Niino; Nakanishi and Niino, 2006) and the RRTMG radiation scheme (rapid radiative transfer model for general circulation models; Iacono et al., 2008). For any other options, please refer to the WRF namelist that is provided as a supplement to this manuscript.*

Line 203: Did you store and use all 33bin of hydrometeor classes to calculate radar observables in CR-SIM?

Yes, we stored and used all bins of all hydrometeor classes to calculate the radar observables for the fast spectral bin simulations. We also stored the aerosol bins (43), but these are not used by CR-SIM. However, the spectral bin scheme uses shared bins for rain / cloud droplets (First 17 bins for cloud droplets, second 16 bins for rain) and cloud ice / snow (First 17 bins for cloud ice, second 16 bins for snow). The output for graupel consists of the full 33 bins. The data is saved at our institute and available on request. We don't think this information is relevant for the reader which is why we did not change the phrasing at this point.

Line 286: "but none of them as pronounced as in the observations." Well, this is typical situations that relatively coarse-resolution model won't be able to resolve tiny cells. So you are running with 2km horizontal grid, meaning that you can resolve convective features in 10km or 20km well, but never be able to resolve 2km-size of convection, which tend to have shallower echo-top heights. So, don't blame to microphysics, but model dynamic core and grid spacing you chose.

This is again connected to a misunderstanding. We use a 400 m horizontal grid spacing. That means we are able to resolve convective cells at 2 km or 4 km in size. However, the point still stands: it is likely that we miss the very small convective cells anyways which correlate to the lower echo-top heights. It was not our intention to blame the cloud microphysics for this, as this is a feature in all simulations independent of the cloud microphysics. We slightly rephrased the sentence to clarify the meaning:

> *All NWP simulations independent of the microphysics scheme are able to reproduce a peak at a similar altitude but none of them as pronounced as in the observations.*

Furthermore, we added another sentence in the following abstract to emphasize that we don't blame the microphysics for this effect:

> *This is independent of the chosen cloud microphysics scheme and mainly a result of the missing small-scale cells in the simulations which is indicative of a resolution effect: the*

*very small cell heights correspond to tiny cells that we might not be able to resolve even
with our 400 m grid spacing.*

Line 313: "Contoured frequency by altitude distributions" -> "Contoured frequency of altitude
diagram"

Changed as suggested, also applied elsewhere in the manuscript.

Line 322: "image 5" -> "Figure 5"?

Changed as suggested.

**References**

Austin, P. M. and Bemis, A. C.: A quantitative study of the "bright band" in radar precipitation
echoes, Journal of Atmospheric Sciences, 7, 145–151, 1950.

Baldauf, M., Seifert, A., Förstner, J., Majewski, D., Raschendorfer, M., and Reinhardt, T.:
Operational convective-scale numerical weather prediction with the COSMO model: Description
and sensitivities, Monthly Weather Review, 139, 3887–3905, 2011.

Brandes, E. A., Zhang, G., and Vivekanandan, J.: Experiments in rainfall estimation with a
polarimetric radar in a subtropical environment, Journal of Applied Meteorology, 41, 674–685,
2002.

Chen, F. and Dudhia, J.: Coupling an advanced land surface–hydrology model with the Penn State–
NCAR MM5 modeling system. Part I: Model implementation and sensitivity, Monthly weather
review, 129, 569–585, 2001.

Ek, M., Mitchell, K., Lin, Y., Rogers, E., Grunmann, P., Koren, V., Gayno, G., and Tarpley, J.:
Implementation of Noah land surface model advances in the National Centers for Environmental
Prediction operational mesoscale Eta model, Journal of Geophysical Research: Atmospheres, 108,
2003

Iacono, M. J., Delamere, J. S., Mlawer, E. J., Shephard, M. W., Clough, S. A., and Collins, W. D.:
Radiative forcing by long-lived greenhouse gases: Calculations with the AER radiative transfer
models, Journal of Geophysical Research: Atmospheres, 113, 2008.

Jung, Y., Xue, M., and Zhang, G.: Simulations of polarimetric radar signatures of a supercell storm
using a two-moment bulk microphysics scheme, Journal of Applied Meteorology and Climatology,
49, 146–163, 2010.

Kumjian, M. R.: The impact of precipitation physical processes on the polarimetric radar variables,
The University of Oklahoma, 2012.

Maxwell-Garnet, J. C.: Colours in metal glasses and in metallic films, Phil. Trans. R. Soc. Lond, A, 203, 385–420, 1904.

Matsui, T., Zeng, X., Tao, W.-K., Masunaga, H., Olson, W. S., and Lang, S.: Evaluation of long-term cloud-resolving model simulations using satellite radiance observations and multifrequency satellite simulators, Journal of Atmospheric and Oceanic Technology, 26, 1261–1274, 2009.

Nakanishi, M. and Niino, H.: An improved Mellor–Yamada level-3 model: Its numerical stability and application to a regional prediction of advection fog, Boundary-Layer Meteorology, 119, 397–407, 2006.

Putnam, B. J., Xue, M., Jung, Y., Zhang, G., and Kong, F.: Simulation of polarimetric radar variables from 2013 CAPS spring experiment storm-scale ensemble forecasts and evaluation of microphysics schemes, Monthly Weather Review, 145, 49–73, 2017.

Ryzhkov, A., Pinsky, M., Pokrovsky, A., and Khain, A.: Polarimetric radar observation operator for a cloud model with spectral microphysics, Journal of Applied Meteorology and Climatology, 50, 873–894, 2011.

Snyder, J. C., Bluestein, H. B., Dawson II, D. T., and Jung, Y.: Simulations of polarimetric, X-band radar signatures in supercells. Part I: Description of experiment and simulated $\rho_{hv}$ rings, Journal of Applied Meteorology and Climatology, 56, 1977–1999, 2017.

---

## Author Comment (AC2)

**Title**: Evaluation of convective cloud microphysics in numerical weather prediction model with dual-wavelength polarimetric radar observations: methods and examples.

**Authors**:     Köcher, Gregor

Zinner, Tobias

Knote, Christoph

Tetoni, Eleni

Ewald, Florian

Hagen, Martin

doi: 10.5194/amt-2021-299

**Summary**: This study compares polarimetric dual-wavelength observations of convective cells observed by three radars to simulations conducted using 5 different microphysics schemes on a large statistical basis. The study is well-motivated and described with clear, informative figures and has the potential to be very well-received as it is a very topical study. The study also benefits from its large sample size (versus individual case studies done in the past). However, in addition to a few changes for clarity and requested further exploration, I have concerns about some of the analysis and conclusions drawn, contributed to by both vagueness of the details of the radar operator and the understood assumptions about the microphysical schemes employed. In addition, some of the analysis seems to rely on simplifying assumptions/conjecture that could potentially be resolved by including additional info from the simulations (e.g., PSDs) besides the bulk polarimetric quantities. Because of the fundamental nature of these concerns, I recommend major revisions before publication in AMT.

Thank you for your detailed explanation of your concerns. You were right about our partial misunderstanding on assumptions on model and forward simulator side in our conclusions. We have re-examined our analysis and updated many parts with the new found understanding. Please see our point-by-point response as below.

**Main comments:**

1. I really think much more information is needed about the forward polarimetric radar operator applied. Even though a citation is given, the realism of the assumptions made about the treatment of 1) particle shapes, 2) particle orientations, and 3) dielectric constants (especially during multi-phase environments, such as melting), etc. could really strongly influence the resultant simulated polarimetric radar variables and thus deserves to be fleshed out here. None of this uncertainty (or the inevitable reduction in variability inherent in applying fixed relations within an operator like this) is currently acknowledged or taken into account in the subsequent analysis. Finally, the lack of details about things like aspect ratio relations provided complicates the understanding of other parts of the discussion, such as

raised in the following comment.

This is a topic that the first reviewer also commented on. We have extended the CR-SIM section with information about the assumptions concerning particle shapes, orientations and dielectric constants in section 2.4. We reevaluated our subsequent analysis to acknowledge the uncertainties as a result of the forward simulator, as well as relate our findings to details of the forward simulator. This includes multiple parts in section 3.2, 3.3, 3.4 and 4. Find below our answer to the first reviewer:

We have not been clear on the assumptions of the radar forward model (CR-SIM). The section of CR-SIM (2.4) has been appended with the assumptions that CR-SIM is making regarding particle shapes, particle orientation and dielectric constants for each microphysics schemes:

> *The dielectric constant of water is 0.92. Solid phase hydrometeors are assumed to be dielectric dry oblate spheriods and are represented as a mixture of air and solid ice. The refractive index hence depends on the hydrometeor density and is computed using the Maxwell-Garnet (1904) mixing formula. There are no mixed phased particles simulated. This means mixed phase radar signatures (for example the "bright band", Austin and Bemis, 1950) will not be reproduced by the simulation. In order to simulate polarimetric radar observables, a radar forward simulator must assume particle shapes and particle orientation. The particle orientation assumptions are the same for all schemes. It is assumed that the particle orientations are 2D Gaussian distributed with zero mean canting angle as in Ryzhkov et al. (2011). The width of the angle distributions is specified for each hydrometeor class: 10° for cloud, rain, and ice and 40° for snow, unrimed ice, partially rimed ice, and graupel. Regarding the shape assumptions, cloud droplets are simulated as spherical (aspect ratio of 1) and raindrops are simulated as oblate spheriods with a changing axis ratio dependent on the drop size according to Brandes et al. (2002) in all schemes. For ice hydrometeor classes, the same aspect ratio assumptions are applied for all schemes except the P3 scheme: cloud ice is assumed as oblate with a fixed aspect ratio of 0.2. Snow is assumed as oblate with a fixed aspect ratio of 0.6. Graupel is assumed to be oblate with an aspect ratio that is changing from 0.8 to 1, dependent on the diameter and according to Ryzhkov et al. (2011):*

> *ar = 1.0 − 0.02       if D < 10 mm ,*
> *ar = 0.8              if D > 10 mm .*

> *The P3 scheme does not provide the standard ice hydrometeor classes. Instead, the aspect ratio of small ice (spherical, fixed aspect ratio of 1), unrimed ice (oblate, fixed aspect ratio of 0.6), partially rimed ice (oblate, fixed aspect ratio of 0.6) and graupel (spherical, fixed aspect ratio of 1) is assumed by CR-SIM. This means in comparison to the other schemes that the P3 simulation deviates for small ice (aspect ratio of 1 in P3, while cloud ice in other schemes is assumed to have an aspect ratio of 0.2) and graupel (0.8 - 1 in other schemes, while graupel particles in P3 are assumed to have an aspect ratio of 1). Resulting differences in the radar signal are discussed in the result section 3 whenever it might influence the simulated radar signal.*

We further added a small abstract in the conclusions acknowledging uncertainties produced by the radar forward operator:

> *Furthermore, there are uncertainties connected to the radar forward simulator applied. To calculate scattering characteristics, assumptions have to be made including the particles aspect ratio, orientations, shape and more. The variability of the simulated signals is reduced by applying fixed relations compared to the potential variability of shapes, orientations and aspect ratios in nature. In addition, the radar forward simulator applied in our study does not consider mixed phase particles. This means that, e.g., effects such as the bright band where particles melt cannot be reproduced by the simulations. To circumvent some ambiguities introduced this way, the comparison could be extended from radar signal space to cloud hydrometeor space. I.e., retrieved hydrometeor classes can be compared to simulated ones.*

Finally, we added multiple parts in the discussion and conclusions that relate forward simulator details to the results:

Section 3.2

> *Given that the forward simulator applied in this study does not consider wet particles, we find the high bias in Z exists even without considering wet graupel and comes mostly from rain, suggesting PSDs that contain too many large rain drops compared to the observations.*

Section 3.3:

> *Furthermore, 3) the observed variability of ZDR is possibly not correctly captured by the radar forward simulator which has to assume fixed distributions of particle orientations as well as a fixed aspect ratio of the particles.*

Section 3.3:

> *All schemes assuming spherical cloud ice or with other dominating spherical hydrometeor classes at these heights show small ZDR. This is true for the P3 small ice fraction for which the forward simulator assumes spherical aspect ratio of 1. In the Thompson schemes, the assumed aspect ratio by the forward simulator is 0.2, suggesting that other hydrometeor classes with lower ZDR like snow or graupel dominate the signal. Only for FSBM and Morrison (aspect ratio 0.2) cloud ice dominates the signal. The stronger signal in FSBM and Morrison is not a result of different density assumptions, because both, the FSBM and Morrison scheme assume lower density of cloud ice compared to Thompson. The observations do not show increased ZDR at these heights. This could either mean that 1) there are no large cloud ice particles observed, 2) that the signal is dominated by other more spherical particles in the observations, or 3) that the assumed aspect ratio of 0.2 by the radar forward operator is unrealistic and the observed particles are more spherical in nature.*

Section 4:

*This could either be a result of simulated cloud ice particles being too large or too many, but this could also be a result of the assumed flat cloud ice shape with an aspect ratio of 0.2.*

2. There seems to be some confusion about the nature of the microphysics schemes employed that influences some of the manuscript's analysis and main conclusions. The primary issue is with regard to the Thompson microphysics scheme, although similar language/conclusions permeate the paper. The authors state on line 186 that snow is "not considered to be spherical" in Thompson in contrast with other schemes, which treat particles as spherical. An examination of the Thompson et al. (2008) manuscript indeed sees similar language employed to describe the scheme, which has a mass-size exponent that differs from the "spherical" value of 3. This value, of course, comes from the volume of spherical particles ($D^3$) multiplied by a density that varies inversely with diameter ($D^{-1}$), as stated in the abstract of Thompson et al. (2008), leading to an ultimate dependence of mass on $D^2$. However, despite the language used concerning this exponent, which upon reflection I now consider a misnomer, this does not actually ensure that the treatment of the particles is non-spherical in the physical shape sense. In fact, I am not aware of any operational microphysics scheme that actually explicitly predicts the shape of the snow with the exception of the FSBM (and possibly the P3?), that uses fixed aspect ratio-size relations to evolve particle shape as mass is gained/lost (see A1 in Shpund et al. 2019). However, other schemes may implicitly incorporate some shape information through things like the capacitance term in the deposition/sublimation rate equations, etc (for example, this is done in Thompson; see Deposition/sublimation section in the Appendix in Thompson 2008), but it isn't clear that this implicit information is actually being used by the radar operator. Hence, similar language about other schemes (e.g., Line 221 about "non-spherical" snow in the P3) is also potentially misleading.

*In fact, the shape assumptions are applied in the forward simulator and this has to be made clear. We have added this information as in our answer to major point 1. We then carefully reevaluated our discussion and conclusions to remove the misleading language.*

This confusion in framing/treatment leads to incorrect assumptions further on, such as line 360 where it is stated that the Thompson scheme actually treats snow as "oblate" particles in a way that would actually affect scattering amplitudes at different polarizations. That is, to my understanding, only something that would be specified within the forward polarimetric radar operator, which is why it is important to include details of how shapes, etc. are being handled as per Main Comment 1. One could envision two alternative scenarios in conflict with these ideas: a model scheme that considered snow to be "spherical" (in the Thompson parlance) for microphysical purposes that has a constant density and an m-D relation with an exponent of 3 but that in the radar operator is assigned an aspect ratio < 1 that results a ZDR > 0 dB. Alternatively, one could have an m-D relation with an exponent of 2 that was "nonspherical" (in the Thompson parlance) but that in the radar operator

treated all snow as spheres regardless of the density varying across the size spectrum, resulting in a ZDR of 0 dB regardless.

As a result, the assertion on line 359 that the particles are being treated as "spherical" in the FSBM scheme is the reason for its poor ZDR observational agreement is (to my understanding) necessarily incorrect, as 1) shape *is* predicted in the FSBM via size-shape equations, 2) the inverse-dependence of density on particle diameter for snow is also taken into account in the FSBM, so it is "non-spherical" even in the $D^2$/Thompson parlance, but also 3) this also leads to the incorrect conclusion that that is related to why the simulated ZDR is 0 with no mention of the radar operator. If snow and ice particles were in fact treated as spheres in the radar operator, where the ZDR calculations are actually being performed, every single ZDR value aloft would be 0 dB, but we do see spread apparent in the CFADs even in the FSBM and Morrison schemes, which can't be explained by this theory of spherical treatment in the microphysics scheme. All of this also obfuscates the role that density and the assumed PSD form in each scheme are likely playing in the spread of ZDR values aloft, which are hardly discussed at all in the manuscript's results section. I believe much of the analysis needs to be re-examined in light of these understandings.

The concerns connected to our conclusions, especially regarding the ZDR signals are clearly justified and we understand that we misunderstood the role of the microphysics schemes in contrast to the role of the radar forward simulator. We reevaluated our discussion and our conclusions in light of the new understanding.  In general, we now attribute the differences between the schemes more clearly to differences in the underlying particle size distributions or to density assumptions of the radar forward operator. Major changes are made to the discussion and conclusion in section 3.2, 3.3, 3.4 and 4. Please see our marked-up version for the details and our answers to the corresponding specific comments.

**Specific comments:**

1. Line 105: Were cases chosen in any systematic way (e.g., precipitation intensity, coverage, etc) or just randomly throughout the 2019 and 2020 seasons?

   The days were not chosen systematically. We were measuring when convective precipitation was forecast.

2. Line 115: Is there a reason KDP was neglected in the analysis? It is available from CR-SIM and would provide important additional information for contextualizing the differences between observations and simulations. Otherwise please include an explanation of why the study was limited to Z/ZDR/DWR.

   The main reason is that our KDP observations are very noisy. We added this as an explanation to the manuscript in section 3.3:

   *We found KDP to provide not much additional value, in part due to noisy observations, which is why we neglect KDP in the subsequent analysis.*

Apart from this, we could not find additional information from the KDP CFAD that was not already visible in the ZDR CFAD. See the KDP CFADs below:

[Figure]

*Figure 1: CFADs of simulated and measured specific differential phase over 5 convective days in 2019. Radar observations with Poldirad.*

3. Section 2.1: Can information about any efforts for radar calibration be included, particularly for the non-operational research radars? Poor calibration could, in theory, affect both Z and especially ZDR.

Yes, we added the following passage to section 2.1:

*The absolute calibration of reflectivity Z of Poldirad is estimated to have an error of ± 0.5 dB from calibration with an external electronic calibration device (Reimann, 2013) while the reflectivity error of Mira-35 is estimated to be ± 1.0 dB (Ewald et al., 2019). We estimate Poldirad ZDR to have an offset of about 0.15 dB from measurements in a liquid cloud layer where ZDR near 0 is to be expected. This offset is corrected before any of the subsequent analysis is done.*

The ZDR offset was not corrected in the previous version, we corrected this for the current version of the manuscript.

4. Line 132: It isn't clear how far away from the radar these cells typically were when being scanned, but was any effort made to correct the ZDR values to account for high-elevation scans in the RHIs?

We assume this comment refers to the different viewing angle towards particles when doing high-elevation scans instead of low-elevation scans (The beam illuminates particles from below, instead of the side which results in different ZDR values). The CR-SIM radar simulator is able to simulate the radar position and simulates the viewing geometry, so this is accounted for in our simulations. It should be noted that in the vast majority of our cases, we looked at the convective cells from the side, i.e., the Poldirad elevation angle was never above 40°.

5. Line 138: Was the cell movement just a simple extrapolation of storm centroids, or done by visual inspection?

Yes, it was a simple extrapolation. We added this information to section 2.1:

*This cell movement is projected using two previous Poldirad overview PPI scans by calculating the displacement at which the cross-correlation between the two PPI images is at maximum.*

6. Line 155: Please include UTC conversions and list in terms of LST instead of 'am/pm'.

We were actually referring to UTC (am/pm was misleading). We clarified this in the manuscript:

*The spin-up always starts at 18 UTC (20 LST) on the previous day. Thus, the 24 hour forecast exactly covers the day of interest (0 - 24 UTC).*

7. Line 156: What exactly is meant by 'nudging' from the GFS? Was the entire (large-domain) model background replaced with the new GFS analysis, or was it incorporated/assimilated somehow? Or did this only apply to the boundary conditions?

By nudging, we mean the WRF option for grid analysis nudging which appends a nudging term to the prognostic equations of temperature, humidity and wind that "nudges" the WRF grid value towards the GFS grid value. This is done for all grid points of the large-domain but it is not a replacement, because the nudging term is much smaller than the physical WRF terms (Nudging coefficient of 0.0006 s$^{-1}$ in our model runs, see namelist.wrf in supplement). Nudging in the planetary boundary layer was turned off, it was applied only above the PBL. We added more detail to the manuscript:

*The parent Europe domain is nudged towards the global GFS data, by appending a nudging term to the prognostic equations for humidity, temperature and wind that*

*"nudges" the WRF grid value towards the closest GFS grid value for each grid point of the Europe domain above the planetary boundary layer (grid analysis nudging).*

8. Line 168: It may help to specify which D is being referred to: maximum diameter, equivolume diameter, etc.

   It is referring to the maximum diameter. This is now specified.

9. Line 179: Just for clarity, I would add "fixed" before non-zero mu just to make clear it is not a free parameter.

   Added as suggested.

10. Line 238: How was this 32 dBZ threshold chosen and why? Is this the default TINT value?

    32 dBZ is a threshold at a common magnitude to identify convective storms. See for example: Dixon and Wiener (1993), 35 dBZ; Muñoz et al. (2018), 35 and 40 dBZ; Han et al. (2009), 35 and 40 dBZ; Jung and Lee (2015), 35 dBZ; Kober and Tafferner (2009), 37 dBZ; Johnson et al. (1998), 30, 35, 40, 45, 50, 55, 60 dBZ. We added the following part to section 2.5:

    *32 dBZ is at a common magnitude to identify convective storms (e.g., Dixon and Wiener, 1993; Jung and Lee, 2015). Higher thresholds potentially miss moderate or weaker convective cells, while lower thresholds will misidentify more non-convective echos as convective cells.*

11. Line 254: When there are multiple Cartesian input grid points for a given target spherical grid, how are they "all included"? Means? Median? Weights? Etc.

    If multiple grid points fall into the same spherical grid, they are weighted by the distance to the radar volume center. We appended the sentence in the manuscript section 2.6 with this information:

    *I.e., for the interpolation to a grid point of the target spherical grid, all Cartesian input grid points that are within the beam width are included with a weight depending on the distance to the radar volume center.*

12. Line 284: It isn't clear to me exactly why 32 dBZ is used as a threshold here – it seems much larger than most studies. I understand 32 dBZ is used for TINT (as per comment 10),

but once a cell is identified can't a lower Z threshold be chosen for cloud/echo top height? Similarly, I am not sure I like the use of 'echo top' here given such a high reflectivity threshold, as this normally applies to thresholds like -10 dBZ or 0 dBZ while 32 dBZ is solidly in the middle of many convective cells. By using "echo top height", it implies something about the depth of the simulated storms, while in actuality the trends seen seem to just reflect high biases in the simulated Z throughout the depth of the cells. Consider alternate language.

The 32 dBZ threshold comes from the TINT cell-tracking and is at a common or even slightly lower value than used in most studies (see answer to specific comment nr. 10). The height of the identified convective core is an output of TINT and hence very straightforward to analyze. That's why we prefer to use the 32 dBZ height instead of using a lower threshold which we would have to do manually after applying the TINT cell tracking. However, we understand that the use of 'cloud top height' is a misleading language, as we use the height of the cell core (or 32 dBZ echo top height) in reality, which is of course not the cloud top height. We changed the language to avoid the use of 'cloud top height' and instead use 'cell core height' or '32 dBZ echo top height'. This was applied throughout the manuscript at multiple occasions.

13. Line 305: This entire paragraph seemed a bit random and out of place to me. The results here are never compared to the findings of Caine et al. (2013), and the subsequent defense of the study has already been thoroughly provided earlier in the paper.

Caine et al. (2013) was using a similar method to compare convective cell characteristics in NWP model and radar observation. That's why we think it is worth to discuss. However, the paragraph is indeed a bit out of place and we never discuss their findings. We moved the paragraph to the actual comparison of the cell geometry, which was done in the two paragraphs before. Furthermore, we removed the part were we defend our study and instead compare our results against theirs (section 3.1):

> *A similar approach to compare cloud geometry in simulation and radar observation was followed in Caine et al. (2013). They objectively compare simulated cell characteristics with observations over 4.5 days after applying a cell-tracking algorithm on their data. Among other things, they found the simulated convective cells to reach higher altitudes on average compared to their radar observations, which is also visible in our analysis. This is independent of the chosen cloud microphysics scheme and mainly a result of the missing small-scale cells in the simulations which is indicative of a resolution effect: the very small cell heights correspond to tiny cells that we might not be able to resolve even with our 400 m grid spacing.*

14. Lines 319-324: It still isn't clear to me if a bias may be being introduced here due to the RHI scanning strategy. Were the +/- 2 deg RHI scans typically still within the precipitation core or on the edges/ flanks of the cells? With all simulated columns being included in the

CFADs it almost seems inevitable that more weak precipitation regions would be captured that way?

The cells typically were not very far away (always closer than 24 km to Mira35), so even with the +/- 2 deg we were typically still within the precipitation core. However, we also analyzed the model data with the center profile only, see below. The model data is then much more noisy, due to the smaller amount of data points. This demonstrates, however, that the differences between radar and model could be in part a result of the radar noise, which is why we added a paragraph in section 4 to acknowledge this uncertainty:

> *Finally, there is more noise in our radar statistics compared to the simulation statistics (for example Figure 5) due to the lower number of data points available from the observations. This could partially explain biases between model and radar, reminiscent of the large observational effort to statistically compare convective cloud characteristics.*

[Figure]

15. Line 332: I know this probably varied among cases but including an approximate ML height here may be useful.

We added this information:

> *While most schemes exhibit a smooth transition from ice to liquid phase, the prominent exception is the P3 scheme for which reflectivities abruptly increase by*

*about 15 dBZ at the melting layer height (approximately at 3.6 km height, varies among cases)*

16. Line 335: While the distributions are certainly broader and extend to higher Z values above the ML, the medians for most schemes still appear quite close to the median in the observations to me.

    At that point, we were referring to the higher Z values that most schemes extend to above the melting layer height. We rephrased the sentence for clarification (section 3.2):

    *Most other schemes directly above the melting layer height extend to higher reflectivities, showing reflectivities greater than 25 dBZ too often.*

17. Line 338: Were PSDs actually examined? It says "(not shown)", but this may be helpful to include. While it is definitely plausible that the graupel produced is too large, could it also be that there's just too much riming in general (so the particle density is the problem, not its size)?

    We were referring to the high reflectivity above the melting layer height which we analyzed by looking at reflectivity CFADs from the single hydrometeor classes. These have shown that the high reflectivity areas above the melting layer height correspond to graupel. We added an Appendix B where we added the corresponding HM-class CFADs, together with this text passage:

    *CR-SIM calculates radar signals for the single hydrometeor classes independently, next to the total signal of all hydrometeors together. Below are CFADs of the signals calculated from the most interesting hydrometeor classes. The CFADs are shown on the original WRF grid and without attenuation correction. The FSBM simulation sometimes showed spurious rain signals at the highest levels (> 10 km). Sometimes there are small numbers of rain drops are present in the largest bins, even though the mixing ratio of rain is 0 in the FSBM simulation. We consider this an error with no physical meaning.*

[Figure]

*Figure 2: CFADs of simulated reflectivity of the graupel hydrometeor class over 5 convective days in 2019.*

[Figure]

*Figure 3: CFADs of simulated reflectivity of the rain hydrometeor class over 5 convective days in 2019.*

However, it is true that our conclusion that graupel must be too large is not necessarily correct: the high reflectivity could also be a result of too dense graupel particles. With our setup, we are not able to distinguish if the density or the size is the problem, but we reevaluated our conclusions (as for main comment 2) to include the possibility that the density could also be the problem. We reworked the part concerning this comment as follows:

> *These extreme reflectivity values are produced mostly by graupel and to lesser extent by rain (see Appendix B for CFADs of radar signals separated by hydrometeor classes). Compared to our measurements these reflectivities are unrealistically large. A high bias in reflectivity could be produced in principle by three mechanics: the simulated particles are 1) too dense, 2) too many, or 3) too large. The graupel densities assumed by the schemes (and correspondingly in the forward simulator) are 500 kg m$^{-3}$ in the Thompson schemes and 400 kg m$^{-3}$ in the Morrison and FSBM scheme. The higher graupel density could explain the higher bias seen in the Thompson scheme compared to the moderate bias in Morrison and FSBM, but the underlying PSD could also play a role.*

18. Line 355: While I have no doubt in general that the flexibility of the FSBM is aiding its ability to reproduce realistic ZDR values, have other factors been considered, such as the different treatment of drop breakup among schemes, etc?

    We have not considered other factors and added this as an information to the manuscript (section 3.3):

    > *However, contributions by other microphysical processes, such as drop breakup or evaporation could also facilitate the ZDR signatures and were not examined in this study.*

19. Lines 357-362: In addition to the issues raised in the main comments regarding lines 359-360, how much of the differences (for example, the narrowness of the ZDR distributions) between schemes is due to differences in the calculated differential attenuation versus differences in 'intrinsic' variables that affect ZDR, such as shape and density? In general the differential impact on density needs to be explored. The profound differences in ZDR at high altitudes between the FSBM/Morrison scheme and the Thompson/P3 schemes also deserves to be explored.

    The calculated differential attenuation does not have a major impact. See below the CFADs with and without simulated attenuation:

[Figure]

*Figure 4:*
*CFADs of simulated and measured differential reflectivity over 5 convective days in 2019. Observation with the Poldirad radar.*

[Figure]

*Figure 5:*
*CFADs of simulated and measured differential reflectivity over 5 convective days in 2019 without simulated attenuation correction. Observation with the Poldirad radar.*

Regarding the profound differences at higher altitudes, we added a text passage in section 3.3 together with the cloud ice CFAD in the appendix:

*At upper levels clear differences between Morrison/FSBM and Thompson/P3 can be seen. Morrison and the FSBM scheme show ZDR values of up to 4 dB at these heights while the Thompsons and the P3 schemes are close to 0 dB. Here, the high ZDRare caused by cloud ice (see Appendix B for CFADs of radar signals separated by hydrometeor classes). All schemes assuming spherical cloud ice or with other dominating spherical hydrometeor classes at these heights show small ZDR . This is true for the P3 small ice fraction for which the forward simulator assumes spherical aspect ratio of 1. In the Thompson schemes, the assumed aspect ratio by the forward simulator is 0.2, suggesting that other hydrometeor classes with lower ZDR like snow or graupel dominate the signal. Only for FSBM and Morrison (aspect ratio 0.2) cloud ice dominates the signal. The stronger signal in FSBM and Morrison is not a result of different density assumptions, because both, the FSBM and Morrison scheme assume lower density of cloud ice compared to Thompson. The observations do not show increased Z DR at these heights. This could either mean that 1) there are no large cloud ice particles observed, 2) that the signal is dominated by other more spherical particles in the observations, or 3) that the assumed aspect ratio of 0.2 by the radar forward operator is unrealistic and the observed particles are more spherical in nature.*

[Figure]

*Figure 6: CFADs of simulated differential reflectivity of the cloud ice hydrometeor class over 5 convective days in 2019. The P3 scheme does not provide the classical cloud ice category.*

20. Line 366: While size is definitely the main factor, I would not diminish the role that density plays in determining the resonance parameter and thus whether non-Rayleigh scattering is occurring. (Although, of course, in these simplified schemes density is at best a simple function of size, so it isn't a free parameter...)

    We have appended the corresponding sentence in section 3.4:

    *In contrast, DWR is rather sensitive to the particles size. In principle, it is also sensitive to the particle density, but the simulated density is assumed to be constant or a function of particle size.*

21. Line 380: This is incorrect. While the P3 is indeed more flexible, Thompson et al. (2008) says in its abstract, "[this scheme employs] a bulk density that varies inversely with diameter as found in observations and in contrast to nearly all other BMPs."

    This was indeed incorrect. Snow is not of constant density in Thompson. We removed this statement. The following argument about P3 being more flexible still holds, as the P3 scheme uses multiple varying mass-size relations opposed to the one in the Thompson scheme.

22. Line 390: I appreciate the discussion about the potential erroneous growth by collection that is influencing the DWR below the melting layer. However, it is also interesting how different the DWR already is immediately below the ML. It seems to me that the particles leaving the melting layer may be very different in size between the observations and simulations. In the observations, perhaps stochastic breakup is occurring toward the surface that is reducing the DWR of large droplets in the obs while the simulated drops are too small and grow by collection instead? I think this is worthy of further exploration since it is one of the most pronounced differences between observations and simulations.

    The DWR is indeed very different already directly below the ML height. Because the second reviewer was also interested in the profound differences directly below the ML height, we explored this further and calculated rain PSDs.
    Below is our answer to the first reviewers major comment nr. 3):

We provided a CFAD of rain drop size distributions for convective cells (Appendix B):

[Figure]

he following passage was added to section 3.3:

> *In order to separate the analysis into reasons due to differences in the underlying modeled microphysics and due to different processing in the forward simulator, we examined rain particle size distributions directly produced by the NWP model (Rain PSD CFAD in  Appendix B.) The FSBM scheme provides  the drop size distributions over a number of size bins, for the other schemes we calculated the distributions according to the schemes parameterization. Only model grid boxes that were flagged as a convective cell by the TINT cell tracking are considered. The rain PSD CFAD confirms the findings of the ZDR CFAD: the two Thompson schemes simulate large rain drops from the surface up to the melting layer height and even above, while the Morrison scheme produces large rain drops only at the surface and the FSBM produce the highest frequency of small drops.*

Apart from the paragraph to the rain PSDs, we adjusted the DWR discussion corresponding to this comment as follows:

> *Below the melting layer the observed DWR steadily decreases towards the ground. The models do not reproduce this very well: Even though the DWR decreases in all models, this decrease happens abruptly at the melting layer. The DWR directly below the melting layer height is very different between the models and the observations, suggesting that particles falling out of the melting layer are larger in the observations compared to the simulated particles. Below this height the simulated DWR stays more or less constant while the observed decreases towards the surface.*

> *In the P3 simulations (and weaker in the Morrison scheme) the DWR even increases again towards the ground. At these heights rain and graupel are the dominant species. The simulated increase of DWR towards the ground is likely a result of the simulated collection process: Rain droplets grow while falling by collecting smaller droplets. This is visible also directly in the rain PSD (see appendix B) and was discussed in the previous section 3.3. Opposed to this, the large particles precipitating from the melting layer seem to shrink towards the ground, perhaps by drop breakup or evaporation. The general magnitude of simulated DWR near the surface is close to the observed again at around -3 to 10 dB.*

23. Line 392: I am confused by the sudden discussion about vertically pointing radars, which were not used in this study?

    If anything, the discussion about our scanning setup versus typically vertically pointing radars belongs to the introduction. However, after reflecting, we think this is of minor importance, that's why we removed this part completely.

24. I don't think the acronyms need to be redefined in the summary (e.g., NWP, FSBM, PSD, etc).

    We removed the definitions of acronyms in the summary.

25. Line 445: Again, I am not sure this is a correct conclusion to draw as it depends more on the details of the radar forward operator.

    We removed this conclusion, as part of our answer to main comment nr 2.

**Typos, etc.**
1. Line 115: "differential phase" should be "specific differential phase".
   Changed.
2. Line 117: For clarity, "single-polarimetric" should be "single-polarization" (some readers automatically infer multiple polarizations from the term 'polarimetric').
   Changed.
3. Line 147 and elsewhere: "times" should be "x" or "by"
   Changed.
4. Line 218: "Mass size" should be "Mass-size"
   Changed.
5. Line 260: db should be dB
   Changed.
6. Line 256: "cumulated" should be "accumulated"
   Changed.
7. Line 280: "extend" should be "extent"
   Changed.

8. Line 324: "image 5" should be "Figure 5"
   Changed.

**Other author comments:**
We repeated the CR-SIM simulations with P3, because we used a different aspect ratio relation for rain for the P3 simulations in the first version. This did not have a strong effect: the number of convective cells changed slightly (from 4768 to 4758 for all cells and 778 to 780 for cells > 7 km) and there are very slight differences in the P3 CFADs as a result of this compared to the first version.

**References**

Austin, P. M. and Bemis, A. C.: A quantitative study of the "bright band" in radar precipitation echoes, Journal of Atmospheric Sciences, 7, 145–151, 1950.

Brandes, E. A., Zhang, G., and Vivekanandan, J.: Experiments in rainfall estimation with a polarimetric radar in a subtropical environment, Journal of Applied Meteorology, 41, 674–685, 2002.

Caine, S., Lane, T. P., May, P. T., Jakob, C., Siems, S. T., Manton, M. J., and Pinto, J.: Statistical assessment of tropical convection-permitting
model simulations using a cell-tracking algorithm, Monthly Weather Review, 141, 557–581, 2013.

Dixon, M., & Wiener, G. (1993). TITAN: Thunderstorm identification, tracking, analysis, and nowcasting—A radar-based methodology. *Journal of atmospheric and oceanic technology*, *10*(6), 785-797.

Ewald, F., Groß, S., Hagen, M., Hirsch, L., Delanoë, J., & Bauer-Pfundstein, M. (2019). Calibration of a 35 GHz airborne cloud radar: lessons learned and intercomparisons with 94 GHz cloud radars. *Atmospheric Measurement Techniques, 12*(3), 1815-1839.

Han, L., Fu, S., Zhao, L., Zheng, Y., Wang, H., & Lin, Y. (2009). 3D convective storm identification, tracking, and forecasting—An enhanced TITAN algorithm. *Journal of Atmospheric and Oceanic Technology*, *26*(4), 719-732.

Johnson, J. T., MacKeen, P. L., Witt, A., Mitchell, E. D. W., Stumpf, G. J., Eilts, M. D., & Thomas, K. W. (1998). The storm cell identification and tracking algorithm: An enhanced WSR-88D algorithm. *Weather and forecasting, 13*(2), 263-276.

Jung, S. H., & Lee, G. (2015). Radar  based cell tracking with fuzzy logic approach. *Meteorological Applications, 22*(4), 716-730.
Kober, K., & Tafferner, A. (2009). Tracking and nowcasting of convective cells using remote sensing data from radar and satellite. *Meteorologische Zeitschrift, 1*, 75-84.

Maxwell-Garnet, J. C.: Colours in metal glasses and in metallic films, Phil. Trans. R. Soc. Lond, A, 203, 385–420, 1904.

Muñoz, C., Wang, L. P., & Willems, P. (2018). Enhanced object-based tracking algorithm for convective rain storms and cells. *Atmospheric Research, 201*, 144-158.

Reimann, J. (2013). *On fast, polarimetric non-reciprocal calibration and multipolarization measurements on weather radars* (Doctoral dissertation, Technische Universität Chemnitz).

Ryzhkov, A., Pinsky, M., Pokrovsky, A., and Khain, A.: Polarimetric radar observation operator for a cloud model with spectral microphysics, Journal of Applied Meteorology and Climatology, 50, 873–894, 2011.

---

## Referee Report (RR1)

**Title**: Evaluation of convective cloud microphysics in numerical weather prediction model with dual-wavelength polarimetric radar observations: methods and examples

**Authors**: Gregor Köcher, Tobias Zinner, Christoph Knote, Eleni Tetoni, Florian Ewald, and Martin Hagen

**DOI**: amt-2021-299

**Summary:** This study compares polarimetric dual-wavelength observations of convective cells observed by three radars to simulations conducted using 5 different microphysics schemes on a large statistical basis. As before, I believe the paper is a strong one that is highly relevant to where the field is headed on evaluating NWP output with polarimetric radar observations. I am happy to report that the authors have responded thoroughly to the previously raised objections, with the manuscript being significantly improved with the newly added discussion and caveats. There are still a few minor things that I think need to be clarified, but no major scientific issues remain. As such, pending the addressing of the minor issues below, I believe the manuscript is suitable for publication in Atmospheric Measurement Techniques.

**Minor Comments:**
1. Line 207: I am still a bit concerned about the language here of snow not being considered to be "spherical" since I think it will imply something about the physical shape of snow (i.e., aspect ratio) within the Thompson scheme to readers which is not what is meant. However, the following sentences do make this a bit clearer. Can this be changed to something along the lines of, "Snow is not considered to have a constant density across the particle size distribution; rather, the mass is proportional to D^2..."?
2. Lines 241-245: The same concerns exist here as the previous comment. I understand what the authors mean, but still think invoking the word "spherical" will imply something about the shape of the particles when what is really meant is how density varies with size. Please amend these descriptions in an analogous manner to the previous comment. (Note: the discussion appears to have been appropriately modified on lines 440-450).
3. Line 257: Please clarify which species is the matrix and which species is the inclusions within the Maxwell Garnett mixture.
4. Line 324: I am a bit confused – is there anything pertaining to horizontal area of the cores in Figure 4? It appears to just be the height of the top of the 32-dBZ core and the maximum Z within those cores?
5. Line 435: This is a minor point, but should Figure B5 really be described as a CFAD, since it is showing a DSD at each height rather than a 2D histogram of single values by height? In other words, isn't this really more like a mean DSD at each level more

than a CFAD? Or is it actually the relative frequency of occurrence of any number of particles existing with a given size bin, which then naturally emulates a DSD because larger drops are rarer? Given the use of bulk schemes it isn't clear to me how the latter would work (i.e., what cutoff would be used in the PSD to say a drop exists in a bin).

6. Relatedly for Figure B5 (and B3), I am surprised to see any rain at all up to 8-10 km. Are these just within the most powerful updrafts, or are they supercooled water that have somehow reached raindrop sizes? While rare in most schemes and probably more a reflectance of low absolute frequencies, ZDR values for rain of 1-4 dB at 10 km is somewhat surprising (e.g., in the Thompson aerosol-aware and FSBM schemes). (Edited to add: I see now this is discussed on lines 618-620. Despite it being in the Appendix, it may not hurt to move the mention of this potential error to within the main text, although I will leave it up to the authors to decide).

7. Lines 443-445: It appears to me like graupel is dominating the reflectivity at those heights (i.e., Figure B1 vs. B2) rather than rain as stated, since the overall ZDR will be weighted by each species' reflectivity. From the CFADs, the graupel Z is typically on the order of 10-40 dBZ while the rain Z is -15 to 10 dBZ around 4 km AGL. I do think it's true that the sparse appearance of higher ZDR above the melting layer is due to rain rather than graupel, but overall I still think graupel dominates the ZDR signature above the ML. (Edited to add: when this was summarized on line 567, it was clearer that the authors were referring to rain dominating the sparse but large values of ZDR above the ML while graupel dominates the bulk of the signal overall. Perhaps the earlier discussion could be rephrased to make it clearer that the anomalously high ZDR at these levels are predominantly rain rather than overall).

8. Line 539-540: It is interesting that the overall Z is too high in the simulations below the ML which suggests drops that are too large (in agreement with the observed ZDR biases in this region), but the DWR below the melting layer are too low suggesting that raindrops exiting the ML are too small in the simulations. How do the authors reconcile those two results?

**Typos/Grammar/Errata:**
1. Line 52, 54: "super cell" → "supercell"
2. Line 121: "dual-polarimetric" → "polarimetric"
3. Line 257: "Maxwell-Garnet" → "Maxwell Garnett"
4. Line 435: Missing )
5. Line 617: "interesting" → "relevant" or "pertinent"

---

## Author Response (AR2)

Summary:

This study compares polarimetric dual-wavelength observations of convective cells observed by three radars to simulations conducted using 5 different microphysics schemes on a large statistical basis. As before, I believe the paper is a strong one that is highly relevant to where the field is headed on evaluating NWP output with polarimetric radar observations. I am happy to report that the authors have responded thoroughly to the previously raised objections, with the manuscript being significantly improved with the newly added discussion and caveats. There are still a few minor things that I think need to be clarified, but no major scientific issues remain. As such, pending the addressing of the minor issues below, I believe the manuscript is suitable for publication in Atmospheric Measurement Techniques.

Minor Comments:

1. Line 207: I am still a bit concerned about the language here of snow not being considered to be "spherical" since I think it will imply something about the physical shape of snow (i.e., aspect ratio) within the Thompson scheme to readers which is not what is meant. However, the following sentences do make this a bit clearer. Can this be changed to something along the lines of, "Snow is not considered to have a constant density across the particle size distribution; rather, the mass is proportional to D^2…"?

Changed the sentence to the following:

> *Snow is not considered to have a constant density across the particle size distribution, the mass is proportional to D^2 (b=2) to better fit observations.*

2. Lines 241-245: The same concerns exist here as the previous comment. I understand what the authors mean, but still think invoking the word "spherical" will imply something about the shape of the particles when what is really meant is how density varies with size. Please amend these descriptions in an analogous manner to the previous comment. (Note: the discussion appears to have been appropriately modified on lines 440-450).

We changed the phrasing, following a similar language as in Morrison and Milbrandt (2015):
> *Unrimed ice, grown by vapor diffusion or aggregation, and partially rimed ice have an effective density that is generally less than that of an ice sphere (b=1.9). The parameter a follows an empirical relationship from Brown and Francis (1995) (a = 0.0121 kg m$^{-b}$) for unrimed ice and depends on the rime mass fraction $F_r$ for partially rimed ice (a = 0.0121/(1 − $F_r$ ) kg m$^{-b}$), i.e., a increases with the rime mass fraction.*

3. Line 257: Please clarify which species is the matrix and which species is the inclusions within the Maxwell Garnett mixture.

Ice is the matrix, air the inclusion (Oue et al., 2020). Text passage adjusted as follows:

> *Solid phase hydrometeors are assumed to be dielectric dry oblate spheroids and are represented as air in an ice matrix. The refractive index hence depends on the hydrometeor density and is computed using the Maxwell Garnet (1904) mixing formula.*

4. Line 324: I am a bit confused – is there anything pertaining to horizontal area of the cores in Figure 4? It appears to just be the height of the top of the 32-dBZ core and the maximum Z within those cores?

Thank you for this catch. We analyzed the horizontal area in an early version of the manuscript but decided that the cell core height is sufficient for the publication (because it is strongly correlated to the cell area). The reference to the area is a remnant from this early version. We removed it.

5. Line 435: This is a minor point, but should Figure B5 really be described as a CFAD, since it is showing a DSD at each height rather than a 2D histogram of single values by height? In other words, isn't this really more like a mean DSD at each level more than a CFAD? Or is it actually the relative frequency of occurrence of any number of particles existing with a given size bin, which then naturally emulates a DSD because larger drops are rarer? Given the use of bulk schemes it isn't clear to me how the latter would work (i.e., what cutoff would be used in the PSD to say a drop exists in a bin).

It is actually showing the relative frequency of occurrence at a given particle size, so it is indeed a CFAD. Using the bulk-parameterization formulas for the rain PSDs, one can calculate the number of droplets for a given diameter. We calculated the number of droplets for the diameters that correspond to the FSBM bins (more specific: the diameter at the geometric center of each bin). By summing up all time steps and grid boxes, we obtain a total number of drops at each height and for each of the bin center diameters. This is visualized as a relative frequency distribution over the height which is, to our understanding, a CFAD. We clarified this passage in the manuscript:

> *The FSBM scheme provides the drop size distributions over a number of size bins, for the bulk schemes we calculated the distributions according to the schemes parameterization. The FSBM bins are approximated by calculating the number of droplets for the geometric centers of the FSBM bins. The calculated number of droplets for the given bin center diameters are then summed up over all time steps and over the grid boxes at each height and visualized as a relative frequency.*

6. Relatedly for Figure B5 (and B3), I am surprised to see any rain at all up to 8-10 km. Are these just within the most powerful updrafts, or are they supercooled water that have somehow reached raindrop sizes? While rare in most schemes and probably more a reflectance of low absolute frequencies, ZDR values for rain of 1-4 dB at 10 km is somewhat surprising (e.g., in the Thompson aerosol-aware and FSBM schemes). (Edited to add: I see now this is discussed on lines 618-620. Despite it being in the Appendix, it may not hurt to move the mention of this potential error to within the main text, although I will leave it up to the authors to decide).

The radar signals from rain at the very upper heights in the FSBM scheme have pretty clearly no physical meaning, because the actual rain mixing ratio at these grid boxes is 0. That's why we don't think it is worth to discuss in the main text which would, in our opinion, distract from the more

important aspects. This is why we think the appendix is better suited for this discussion and we left it as it is.

7. Lines 443-445: It appears to me like graupel is dominating the reflectivity at those heights (i.e., Figure B1 vs. B2) rather than rain as stated, since the overall ZDR will be weighted by each species' reflectivity. From the CFADs, the graupel Z is typically on the order of 10-40 dBZ while the rain Z is -15 to 10 dBZ around 4 km AGL. I do think it's true that the sparse appearance of higher ZDR above the melting layer is due to rain rather than graupel, but overall I still think graupel dominates the ZDR signature above the ML. (Edited to add: when this was summarized on line 567, it was clearer that the authors were referring to rain dominating the sparse but large values of ZDR above the ML while graupel dominates the bulk of the signal overall. Perhaps the earlier discussion could be rephrased to make it clearer that the anomalously high ZDR at these levels are predominantly rain rather than overall).

Yes, we were referring to the sparse but large ZDR values. These values are of course not dominating the whole signal which is dominated by graupel as correctly stated in this comment. We rephrased this part to remove the misleading phrasing:

> *The signal directly above the melting layer height is generally dominated by graupel, which has the highest reflectivity signal (see Appendix B for separation by hydrometeor class) and is associated with ZDR values of 0, due to the assumed aspect ratio of 1 in the forward simulation. The sparse but large values of ZDR in the two Thompson and the P3 scheme are predominantly caused by rain, likely lifted by strong updrafts in the convective situations.*

8. Line 539-540: It is interesting that the overall Z is too high in the simulations below the ML which suggests drops that are too large (in agreement with the observed ZDR biases in this region), but the DWR below the melting layer are too low suggesting that raindrops exiting the ML are too small in the simulations. How do the authors reconcile those two results?

Comparison of the simulated and observed DWR CFADs must be done cautiously because the first DWR CFAD does not include simulated attenuation while the observed radar signal is naturally attenuated. Including the simulated attenuation increases the simulated DWR by a lot (as shown in the second DWR CFAD). We therefore think the conclusion that simulated rain drops exiting the ML are too small based on the uncorrected DWR CFAD should not be drawn. We changed the corresponding sentence:

> *The DWR directly below the melting layer height is very different between the models and the observations. However, including attenuation increases the simulated DWR and its variability, making it difficult to quantify DWR deviations between model and observations as discussed below.*

Typos/Grammar/Errata:

1. Line 52, 54: "super cell" ﹏ "supercell"
Changed as suggested.

2. Line 121: "dual-polarimetric" ﹏ "polarimetric"
Changed as suggested.

3. Line 257: "Maxwell-Garnet" ﹏ "Maxwell Garnett"
Changed as suggested.

4. Line 435: Missing )
Added.

5. Line 617: "interesting" ﹏ "relevant" or "pertinent"
Replaced "interesting" with "relevant".

**References**

Brown, P. R. and Francis, P. N.: Improved measurements of the ice water content in cirrus using a total-water probe, Journal of Atmospheric and Oceanic Technology, 12, 410–414, 1995.

Maxwell-Garnet, J. C.: Colours in metal glasses and in metallic films, Phil. Trans. R. Soc. Lond, A, 203, 385–420, 1904.

Morrison, H. and Milbrandt, J. A.: Parameterization of cloud microphysics based on the prediction of bulk ice particle properties. Part I: Scheme description and idealized tests, Journal of the Atmospheric Sciences, 72, 287–311, 2015.

Oue, M., Tatarevic, A., Kollias, P., Wang, D., Yu, K., and Vogelmann, A.: The Cloud-resolving model Radar SIMulator (CR-SIM) Version 3.3: description and applications of a virtual observatory, Geoscientific Model Development (Print), 13, 2020.